# Environmental and Economic Assessment of Alternative Food Waste Management Scenarios

Dimitrios Mathioudakis [1], Panagiotis Karageorgis [1], Konstantina Papadopoulou [1,*], Thomas Fruergaard Astrup [2] and Gerasimos Lyberatos [1,3]

1   School of Chemical Engineering, National Technical University of Athens, Iroon Polytechneiou 9, Zografou, 15780 Athens, Greece
2   Department of Environmental and Resource Engineering, Technical University of Denmark, 2800 Kgs Lyngby, Denmark
3   Institute of Chemical Engineering Sciences (ICE-HT), Stadiou Str., Platani, 26504 Patras, Greece
*   Correspondence: kpapado@chemeng.ntua.gr; Tel.: +30-210-772-3115

**Abstract:** The scope of this paper was to examine the environmental and economic performance of alternative household fermentable waste (HFW) management scenarios. In Greece, the business-as-usual scheme for the management of HFW is its disposal in landfills as part of mixed waste. Within a HORIZON2020 called Waste4think a series of alternative approaches based on the benefits of source separation was developed. Specifically, source separated HFW is led to a drying/shredding plant, located in the municipality, for the production of a high-quality biomass product, which is called FORBI (Food Residue Biomass). Alternative approaches have been examined for the exploitation of FORBI: a simple alternative consists of the transportation of food waste (without drying/shredding) to the landfill, composting and covering the landfill's layers with the produced compost. On the other hand, a set of technological alternatives examined are: one- and two-stage anaerobic digestion for the production of biogenic compressed natural gas (bio-CNG) and bio-hythane, composting and utilization of compost in the municipality, bio-ethanol production and pelletization. The alternatives have been assessed using Life Cycle Assessment and Life Cycle Costing tools. The results show that both the simple and the innovative alternatives proposed perform better than the baseline scenario both in economic and environmental terms.

**Keywords:** food waste; LCA; LCC; EASETECH; valorization

## 1. Introduction

Food waste (FW), which corresponds to the largest percentage of municipal solid waste (MSW), typically over 40%, could represent one of the most prominent resources of the future [1,2]. The biochemical composition of FW makes it an excellent source of several valuable compounds (nutrients, chemicals, materials, fuels, etc.) as shown in numerous research articles [3–6]. The potential value from alternative FW valorization approaches, if society moves away from the sub-optimum approaches of incineration and disposal of valuable resources, could be enormous, regarding both the economy and the environment [7]. For example, food waste may serve as feedstock for anaerobic digestion, composting and dark fermentation processes, and thereby contribute with a range of products, such as biomethane, compost and biohydrogen. The environmental and economic performances of these valorization pathways are dependent on a wide range of technological parameters. In order to harvest the full valorization potential, FW processing should be understood as a complex concept integrating several processes and outputs. Thus far, relatively few studies have provided systematic assessments of such alternative valorization options.

Most literature has focused on more "traditional" management of FW (i.e., landfilling when no source separation exists, direct incineration when source separation schemes are

in place), while alternative valorization scenarios have not received similar attention [3,8,9]. Hence, it is of paramount importance to provide a holistic assessment of the management of fermentable household waste, especially regarding the more innovative approaches, that might lead to significant environmental and/or economic benefits through the production of high added-value products and energy carriers compared to the ones most commonly used.

Today, anaerobic digestion for production of biogas, a high-quality energy carrier, is the most common FW valorization approach [10–14]. Anaerobic digestion of food waste, as a valorization pathway, offers a variety of benefits, depending on the pre-treatment of the feedstock, the selection of the most suitable technology, the digestate valorization and the configuration of the digestion process [15]. The environmental and economic footprint of this valorization approach is largely dependent on the process characteristics and post-process valorization of effluents (biogas and digestate).

The two-stage anaerobic digestion process leads to the production of a hydrogen/methane mixture named hythane [16–20]. Hythane is a more efficient and clean energy carrier for vehicle fuel than biomethane, since literature studies have shown that diesel substitution by hythane significantly decreased off-gas emissions and increased vehicle efficiency [21].

A promising alternative is the utilization of food waste for the production of bioethanol, which is then used to environmentally upgrade conventional petrol used in vehicles [7,22]. According to previous LCA studies, bioethanol generation from food waste and utilization as an alternative fuel present environmental benefits especially regarding GHGs emissions (Greenhouse Gases) [23,24]. However, no relevant studies assessing the economic performance of such a fermentable household waste management paradigm were identified in the literature.

Solid fuels in the form of pellets can be generated from food waste after specific pre-treatment [25]. The approach used within the framework of Waste4think was the mechanical production of pellets.

Finally, there is the option to valorize food waste through bio-stabilization for the production of compost, which is then applied to soil, leading to recycling of nutrients [26–28]. However, due to the compositional and physical heterogeneity, food waste requires pre-treatment prior to further valorization. The most common pre-treatment approaches are: thermal, mechanical, acid, base and ultrasound [12,14,29].

As indicated above, food waste is a valuable resource suitable for numerous bio-processes for the production of high-value energy carriers as well as various bio-based products. However, the environmental and climate consequences, as well as the economic sustainability of alternative valorization pathways, are unclear and have not been addressed in detail in the literature. Few studies have assessed the combination of several processes into integrated solutions [30]. To further advance the valorization of food waste and avoid loss of bioresources, further environmental and economic assessments are needed.

The MSW management systems are characterized by high complexity (multiple stages, technological parameters, sensitivities to internal and external distortions, etc.), hence their design, development and implementation analysis, performance monitoring and optimization are not usually straightforward, especially in cases where more sophisticated and multi-level technologies are implemented [31–33]. Life Cycle Assessment (LCA) and Life Cycle Costing (LCC) are commonly accepted to be the most suitable tools to overcome the complexity issues and to conduct a detailed, quantitative performance evaluation of an MSW management framework, including the assessment of environmental and economic impacts that minor or major variations (alternative scenarios) will have in the overall performance of the scheme. Furthermore, a preliminary LCA/LCC study serves the purpose of assessing the priority parameters that need to be monitored during the day to day operation of the system by the waste management operators and decision makers, in order to minimize the environmental and/or economic impacts [34–36].

Reviewing the literature on the topic, it is apparent that there is much research focus both on the development of alternative food waste valorization processes and the assess-

ment of alternative waste management scenarios. However, it was possible to identify three major gaps: (i) there are no relevant studies feeding the LCA and LCC models with actual experimental and site-specific data. Instead, most of the studies use generic global data, thus presenting increased uncertainties; (ii) most of the LCA/LCC studies identify food waste heterogeneity as a major burden for optimized valorization; however, there are many non-pre-treatment options assessed; and (iii) few studies propose specific options for local authorities on developing a fully localized FW valorization approach.

The study is based on data and results from the Waste4think project for the development of a data-driven holistic environmental and economic assessment of potential valorization options for food waste from households. To support further application of the results, the study includes an uncertainty analysis to provide stakeholders (i.e., municipalities and authorities) with the necessary information to reach balanced and well-informed decisions [37,38].

Overall, the current study aims at (i) combining experimental outcomes and the LCA and LCC tools to assess the municipality of Halandri, Attica Region, Greece transition to an innovative FW management paradigm, (ii) expanding its contribution beyond the case study-specific outcomes to the support of policy makers in order to develop a wider understanding of the environmental and economic hotspots for the design of integrated and localized household food waste management systems based on site-specific instead of generic data; and (iii) quantifying the benefits and impacts of implementing a source-separated FW homogenization process, which is experimentally proved to enhance bioprocess efficiencies.

## 2. Methodology

### 2.1. Goal and Scope of the Study

The goal of the study was to assess and compare alternative fermentable household waste valorization scenarios, based on the conclusions derived by the Waste4think project. The EASETECH model was used for the development of both the LCA and the LCC models [39].

The methodology implemented included the following steps:

i.      Baseline data collection: In this stage, all the necessary data regarding the baseline situation were collected. The most important amongst these was a detailed analysis of the composition of MSW in the municipality as well as the performance of the source separation schemes, which was performed as described in Mathioudakis et al., 2021 [1]. Moreover, data regarding the collection and transportation framework, cost data, etc., were also collected.

ii.     Performance of alternative processes: lab and pilot scale experiments were carried out to assess the efficiency and performance of the alternative processes developed: productivity (of, e.g., biogas, bioEtOH, hythane, etc.), technological specifications, etc.

iii.    LCA and LCC models were developed based on the aforementioned data in EASETECH. Uncertainty analysis was carried out to optimize quality of results.

iv.     Results were interpreted, environmental and cost hotspots were identified, and the optimum scenario was determined.

The scope of the current study was to provide stakeholders, and specifically municipality authorities with the suitable background information to conduct well-informed decision-making, in developing an environmentally sound and economically feasible fermentable household waste management system. The functional unit was the collection, transportation, treatment and disposal of 1000 kg of municipal solid waste. The study involved a consequential modelling approach and applied system expansion to address co-functionality of the functional unit. A zero-burden approach was applied regarding waste generation.

The goal and scope of the study necessitates the implementation of a methodological framework of a consequential LCA, in order to include the potentially significant effects on other technoeconomic systems [40–42]. The consequential LCA describes the methods to

model system expansion in the case of influencing background systems, when a significant change occurs in the case study's system boundaries (e.g., substitution of diesel with bio-CNG generated through the anaerobic digestion of food waste). Where marginal Life Cycle Inventory (LCI) data were available, they were used, instead of average data, in order to ensure that market mechanisms are taken into consideration in the interactions among adjoining systems. However, it is apparent that the implementation of such a framework includes various uncertainties, which need to be taken into consideration in interpreting the results.

### 2.2. Life Cycle Costing Methodology

The Life Cycle Costing has been conducted using the same system boundaries as in the Life Cycle Assessment, in order to provide a holistic environmental and economic assessment of the alternative scenarios. The methodology followed the principles described by Martinez-Sanchez et al., while the input data have been extracted through analysis of actual local data, literature review and data from the imported to EASETECH datasets [41]. Only "internal" costs, i.e., monetary costs, were included in the assessment, while "external" costs, which correspond to non-marketed goods and services, were excluded from the study.

The cost model followed a Unit Cost Method approach, as described in a previous study [43]. Specifically, for the implementation of the Unit Cost Method, the overall waste system is divided into separate stages/tasks (e.g., source separation, collection and transportation, valorization, etc.). In the current work, the separate activities correspond to the ones defined in the LCA model. Each task is then attributed with the relevant cost items (either capital expenditure or operational expenditure): machinery, personnel costs, energy costs, etc. Then, the physical (units/to of waste) and economic (€/unit) dimensions of each cost item are defined. For example, the physical parameter for the fuel consumption of the waste collection trucks is Ldiesel/tonne MSW, while the economic parameter is EUR/Ldiesel.

Finally, the LCC of the waste management system is estimated as the sum of all the costs associated with the relevant activities of a scenario (Equation (1)):

$$LCC = \sum_{i=1}^{n} [W_i \times (UBC_i + UT_i)] \tag{1}$$

where:

- $n$, the number of activities involved in the scenario;
- $UBC_i$, Unit Budget Cost of activity $i$;
- $W_i$, the amount of waste input into the same activity $i$.

Capital expenditures (CAPEX) are allocated equally to all tonnes of waste inputs to the system throughout the economic lifetime of the corresponding technologies implemented. This approach necessitates the implementation of a discounting methodology. Specifically, lump-sums are converted into annuities using the following Equation (2):

$$A = \frac{P}{\frac{(1+ir)^n}{ir*(1+ir)^n}} \tag{2}$$

where:

- $P$, the present value of each piece of equipment or technology.
- $ir$, the interest rate. In the current study an interest rate of 3.5% was assumed.
- $n$, the economic lifetime of the technology/piece of equipment.

In order to estimate the UBC, each annuity is divided with the annual usage rate of the corresponding technology/equipment. The annual usage rate might be either the annual capacity of the technology or a fraction thereof, if it operates below its maximum capacity.

The operational expenditure (OPEX) is either fixed on an annual basis (e.g., labor, maintenance and insurance) or variable, depending on the scenario's parameters (e.g., energy

consumption). As for CAPEX, OPEX is estimated on a per ton of waste basis, using the following Equations (3) and (4):

$$CIf = \frac{AC}{AUR_p} \tag{3}$$

where:

- CIf, the fixed costs per item for treating one tonne of waste;
- AC, the annual fixed costs for the specific cost item;
- $AUR_p$, the annual usage rate of the plant;

and:

$$CIv = IxT \times UPI \tag{4}$$

where:

- $CI_v$, the variable costs per item for treating one tonne of waste;
- IxT, item's quantity for managing one tonne of waste;
- UPI, unit price of the item.

### 2.3. Impact Categories

The results section provides an overview of the LCA and LCC results for selected impact categories. Selection of impact categories to be presented has been made based on the following criteria [44–48]:

i. Climate Change (CC) represents a global environmental issue that attracts the attention of the global public opinion, and tackling it is the epicenter of the decision-making around the world.

ii. Ozone Depletion (OD) follows the same trend as Photochemical Ozone Formation, Marine Eutrophication and Depletion of Abiotic Resources (elements).

iii. Human toxicity (non-cancer effects) (HT) follows the same trend as Ionising Radiation, Freshwater Eutrophication, Freshwater Ecotoxicity and Depletion of Abiotic Resources (fossil).

iv. Particulate Matter (PM) follows the same trend as Terrestrial Acidification, Human Toxicity (cancer effects) and Terrestrial Eutrophication.

v. Results of the Life Cycle Costing (LCC) are being presented per scenario and in parallel with the Life Cycle Impact Assessment (LCIA).

The International Life Cycle Database (ILCD) method has been used for the calculation of the various impact categories.

### 2.4. Inventory Data

The basis for the alternative scenarios developed within the framework of Waste4think is that source-separated and collected HFW is transferred for drying and shredding in a specially designed area, locally established within the boundaries of the municipality. The material is heat-treated (92–98 °C) and shredded in a GAIA GC-300 dryer/shredder, in order to generate a valuable biomass product named FORBI.

The treatment takes 9 h at 92–98 °C and 2 h of cooling with a maximum power usage of 23.8 kW. The exact processing time and energy use vary somewhat, depending mainly on the original waste moisture content. The drying system reduces the amount of fermentable waste percentage by up to 75%. The moisture content in the household fermentable waste is high (60–80%). The drying/shredding benefits are the reduction of the final volume and weight (thus decreasing storing and transporting costs), the removal of odors and a longer shelf life and product homogeneity.

A summary of the alternative scenarios studies is presented in Table 1:

**Table 1.** Introduction to alternative food waste management scenarios.

| Scenario 0 | Current Situation |
|---|---|
| Scenario 0.1 | Food waste source separation and composting without drying/shredding |
| Scenario 1 | Food waste source separation and composting without drying/shredding |
| Scenario 1.1 | Bio-CNG generation and co-composting with green waste |
| Scenario 1.2 | Hythane production through two-stage anaerobic digestion |
| Scenario 2 | Green-waste and FORBI co-composting |
| Scenario 3 | Bio-EtOH |
| Scenario 4 | Pellets |

2.4.1. Scenario 0, Current Situation

The baseline scenario refers to the current food waste management scheme implemented in the Municipality of Halandri (Figure 1).

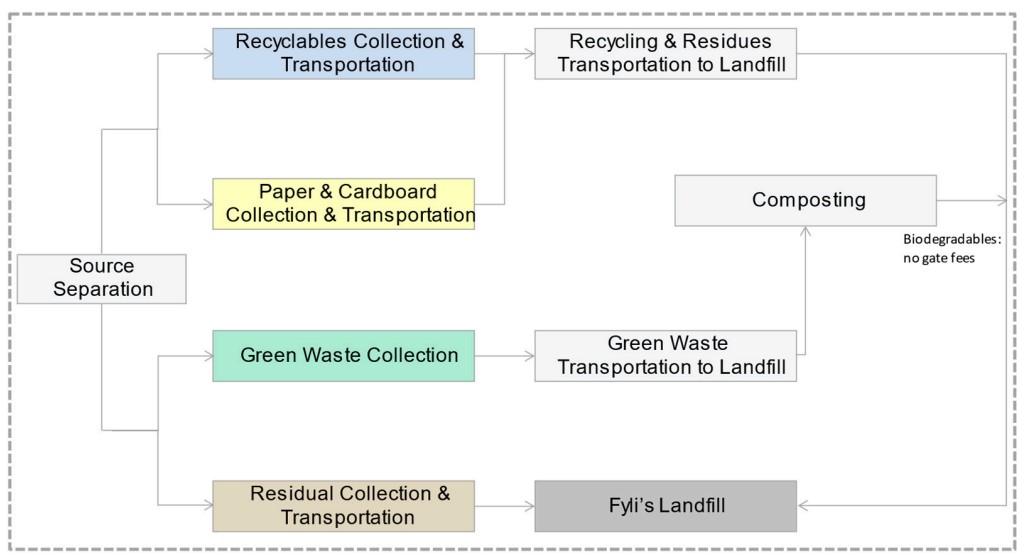

**Figure 1.** Scenario 0, current situation.

Specifically, food waste is currently not in a source separation scheme; instead, it is collected as commingled waste and then transported to a landfill (Fyli's Landfill), which is located in an area approximately 25 km from Halandri. The fuel consumption for the collection and transportation of commingled waste has been estimated to be 0.0024 and 0.00008 L diesel per kg total wet weight, respectively. Concerning the disposal of waste in Fyli Landfill, there are gate fees equal to tonne EUR 53 per tonne (Table 2).

**Table 2.** Inventory table, Scenario 0.

| Input/Output | Value |
|---|---|
| Waste Transportation to Landfill (km) | 25 |
| Fuel Consumption for collection (L/kg total wet weight) | $2.4 \times 10^{-3}$ |
| Fuel Consumption for transportation (L/kg total wet weight) | $8 \times 10^{-5}$ |
| Landfill gate fees (EUR/tonne) | 53 |
| Green waste collection and transportation (EUR/tonne) | 16 |

Green waste (consisting of prunnings, leaves and soil) is source-separated, collected and transported to an area near the landfill, where it is being composted. The compost is being used as landfill coverage, mitigating odors and leakage of landfill gas. The services that are responsible for the collection and transportation of green waste charge an amount of EUR 16 per tonne of collected waste.

### 2.4.2. Scenario 0.1, Food Waste Source Separation and Composting without Drying/Shredding

This scenario was examined to compare a simple and basic alternative to the commingled waste Scenario 0 alternative (Figure 2).

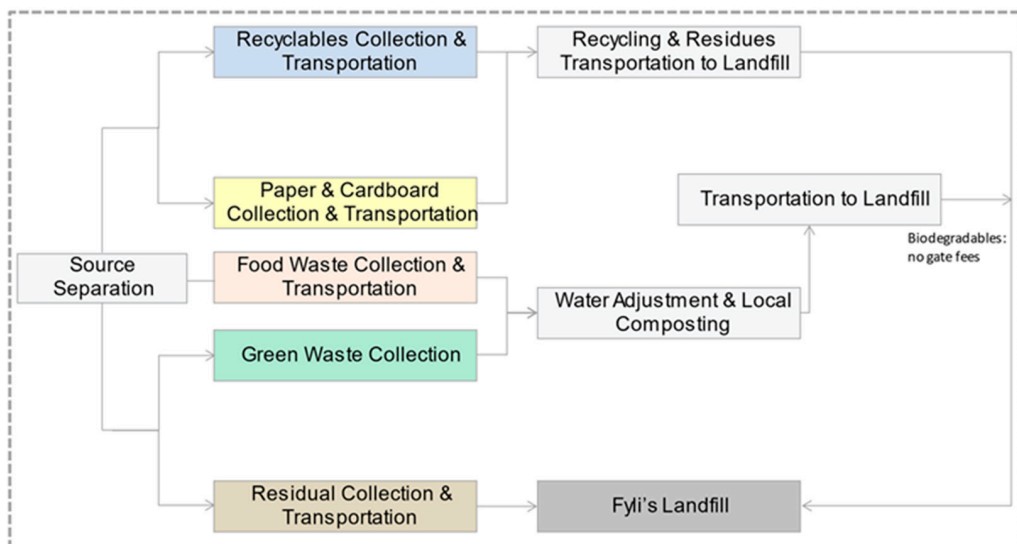

**Figure 2.** Scenario 0.1, food waste source separation and composting without drying/shredding.

Specifically, it is assumed that source-separated food waste is transferred to an area nearby the landfill, where it is being composted alongside green waste. The energy consumption for the composting process was estimated based on literature assumptions to be 0.0436 kWh/kg of input (based on the Greek electricity mix from Ecoinvent). Moreover, there is a consumption of 0.001 L diesel/kg waste for the necessary machinery of the plant (Table 3).

**Table 3.** Inventory table, Scenario 0.1.

| Input/Output | Value |
|---|---|
| Waste Transportation to Landfill (km) | 25 |
| Fuel Consumption for collection (L/kg total wet weight) | $2.4 \times 10^{-3}$ |
| Fuel Consumption for transportation (L/kg total wet weight) | $8 \times 10^{-5}$ |
| Landfill gate fees (EUR/tonne) | 53 |
| Landfill gate fees for green waste (EUR/tonne) | 0 |
| Green waste collection and transportation (EUR/tonne) | 16 |

The compost that is generated is being used to cover landfill layers, leading to carbon sequestration, which generates net environmental benefits.

### 2.4.3. Scenario 1.1, Bio-CNG Generation and Co-Composting with Green Waste

The scenario contains the following steps (Figure 3):

1. Food waste and green waste source-separation;
2. Food waste led to a local treatment facility where it undergoes a drying/shredding process to produce FORBI;
3. A suspension of FORBI is used as a feedstock for an anaerobic digestion process to produce biogas;
4. Biogas undergoes an upgrade process (to remove the rest of the substances included in biogas: moisture, $CO_2$ and $H_2S$) and compressed to generate bio-CNG;
5. Bio-CNG will be used as an alternative fuel for the food waste collection trucks substituting diesel, leading to a corresponding avoidance benefit;

6. The digestate generated is co-composted with green waste (in the suitable ratio), which is an approach that is proven to enhance green waste composting processes.

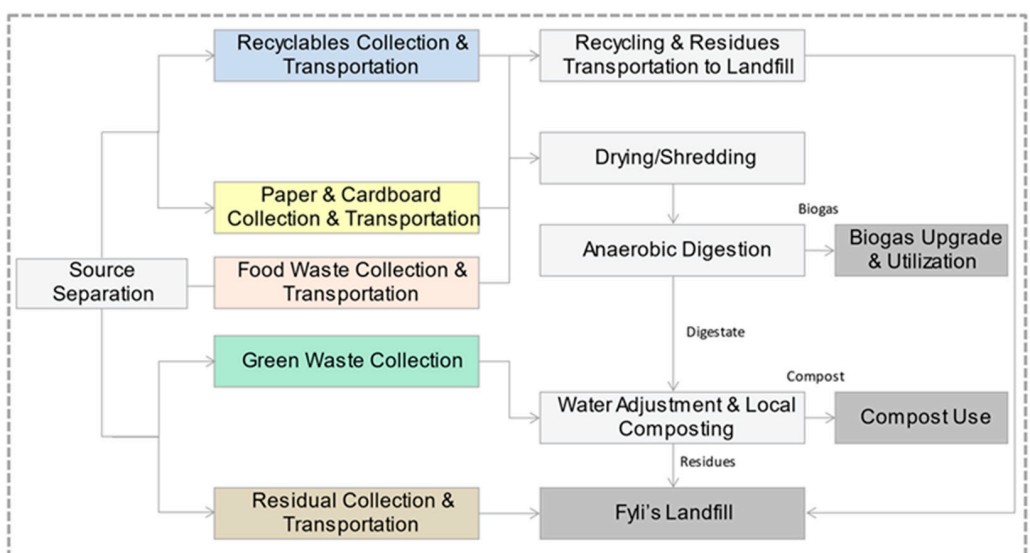

**Figure 3.** Scenario 1.1, bio-CNG generation and co-composting with green waste.

Critical parameters concerning the modelling of anaerobic digestion are the biodegradable carbon that is transformed into methane (equal to 90%) and the measured methane in biogas (65.5%) (Table 3). Energy inputs of the anaerobic digestion include diesel consumption for the necessary machinery at the plant—which equals to 0.0009 L per kg total wet weight input—electricity for the pumps and ventilator operation with a maximum power usage of 0.049 kWh per kg input and energy to heat up water and solids to 35 °C, which equals to 0.0678 and 0.0086 MJ per kg total wet weight, respectively. The generated biogas is then used as a fuel for the waste collection trucks. However, to be suitable for this kind of use, an upgrading and a compression process are necessary. The upgrading of biogas is conducted with the use of methanolamine (MEA) as a $CO_2$ and $H_2S$ adsorbent and silica gel as a moisture adsorbent. There is a maximum power requirement of 0.1 kWh per $m^3$ methane for the regeneration of MEA, which was estimated based on experimental measurements, while for the compression of the upgraded gas, there is an electricity requirement equal to 0.43 kWh per $m^3$ based on the compression equipment's (BRC FMQ 2.5 fuel-maker) specifications. The disposal/post-treatment of MEA has not been included in the assessment system boundaries due to lack of experimental data on the processes that would be necessary for this.

2.4.4. Scenario 1.2, Hythane Production through Two-Stage Anaerobic Digestion

Scenario 1.2 is the same as Scenario 1.1 with the difference that a two-stage anaerobic digestion process is included instead (Figure 4).

Lab and pilot scale experimental data are used as inputs. Hythane, comprised of $CH_4$ 90% and $H_2$ 10%, is generated (Table 4).

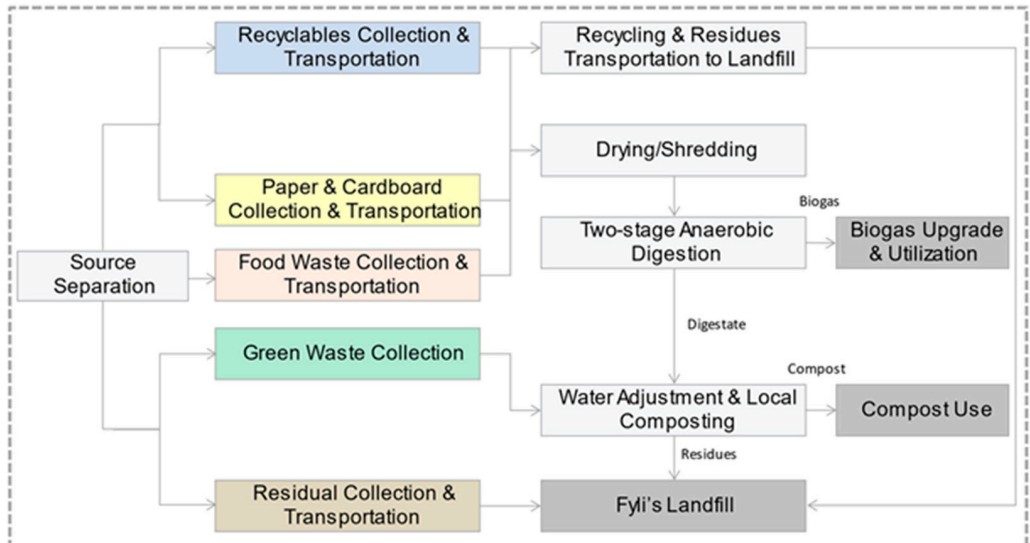

**Figure 4.** Scenario 1.2, hythane production through two-stage anaerobic digestion.

**Table 4.** Inventory table, Scenarios 1.1 and 1.2.

| Input/Output | Value |
|---|---|
| Waste Transportation to Landfill (km) | 25 |
| Fuel Consumption for collection (L/kg total wet weight) | $2.4 \times 10^{-3}$ |
| Fuel Consumption for transportation (L/kg total wet weight) | $8 \times 10^{-5}$ |
| Composting process energy consumption (kWh/kg) | $4.36 \times 10^{-2}$ |
| Composting machinery fuel consumption (Ldiesel/kgfeedstock) | $1 \times 10^{-3}$ |
| Landfill gate fees (EUR/tonne) | 53 |
| Landfill gate fees for green waste (EUR/tonne) | 0 |
| Green waste collection and transportation (EUR/tonne) | 16 |
| AD machinery fuel consumption (Ldiesel/kgtotal wet weight input) | $9 \times 10^{-4}$ |
| Electricity for pumps and ventilator operation (kWh/kgtotal wet weight input) | $4.9 \times 10^{-2}$ |
| Energy to heat up water to 35 °C (MJ/kgtotal wet weight input) | $6.78 \times 10^{-2}$ |
| Energy to heat up solids to 35 °C (MJ/kgtotal wet weight input) | $8.6 \times 10^{-3}$ |
| Maximum power requirement for MEA regeneration (kWh/m3methane) | $1 \times 10^{-1}$ |
| Biogas compression energy consumption (kWh/m3methane) | $4.3 \times 10^{-1}$ |
| FORBI production (tonne/tonneFU) | 0.08 |
| Compost production (tonnecompost/tonneFU) | 0.5 |
| Biogas production (m$^3$/tonneFORBI) | 543 |
| Hythane (90% methane–10% hydrogen) production (m$^3$/tonneFORBI) | 556 |

### 2.4.5. Scenario 2, Green-Waste and FORBI Co-Composting

Pilot-scale experiments have proven that co-composting of municipal green waste with FORBI enhances the efficiency of the process, reducing the composting time to almost half (Figure 5).

Hence, a scenario has been developed including the establishment of a local composting plant within the boundaries of the municipality (either in vessel composting or windrows). In this scenario, the total quantity of FORBI is led to a co-composting process alongside the municipality's green waste (Table 5).

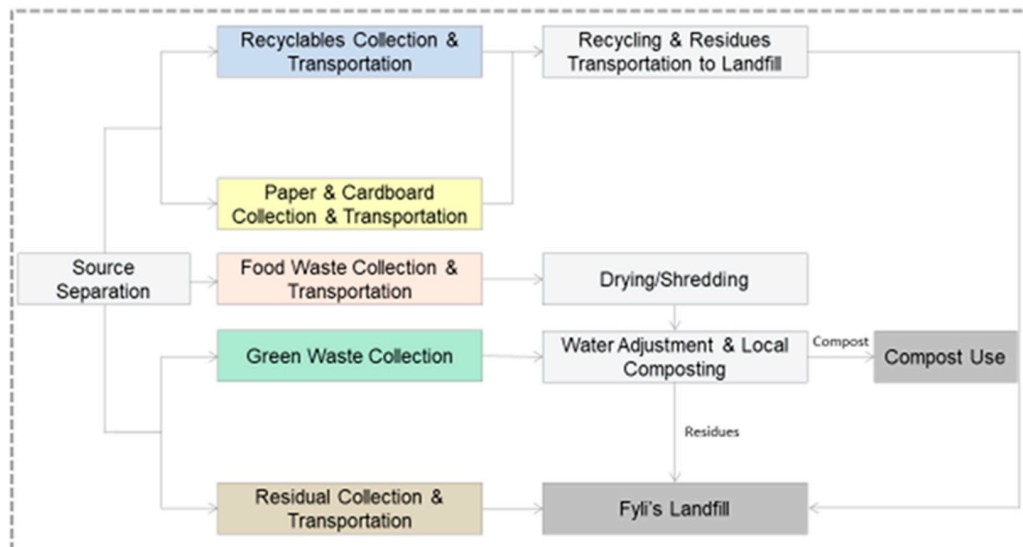

**Figure 5.** Scenario 2, green-waste and FORBI co-composting.

**Table 5.** Inventory table, Scenario 2.

| Input/Output | Value |
|---|---|
| Waste Transportation to Landfill (km) | 25 |
| Fuel Consumption for collection (L/kg total wet weight) | $2.4 \times 10^{-3}$ |
| Fuel Consumption for transportation (L/kg total wet weight) | $5 \times 10^{-5}$ |
| Landfill gate fees (EUR/tonne) | 53 |
| Landfill gate fees for green waste (EUR/tonne) | 0 |
| Green waste collection and transportation (EUR/tonne) | 16 |
| Compost production (tonne/tonneFU) | 0.5 |

The generated compost is used as biofertilizer in the municipality's gardens and parks, substituting chemical fertilizers.

### 2.4.6. Scenario 3, Bio-EtOH

Lab-scale experiments have been pursued to explore the potential of bioethanol generation using FORBI as a feedstock (Figure 6).

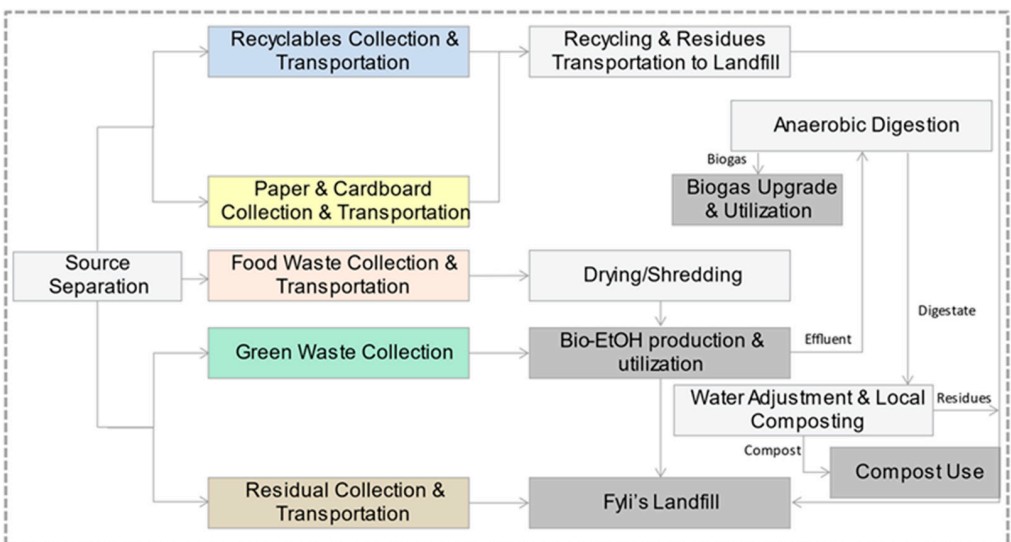

**Figure 6.** Scenario 3, bio-EtOH.

Hence, in Scenario 3, these data will be used to explore options for the utilization of bioethanol as a vehicle fuel. Furthermore, the residue of the bioethanol production process is further treated through anaerobic digestion and the generated digestate through composting.

Raw materials required are sulfuric acid (0.02 kg/kg FORBI) for the pre-treatment of the feedstock and enzymes (0.009 kg/kg FORBI) for the fermentation. Heat and electricity consumption for the bioethanol generation is 4.1 MJ/kg$_{input}$ and 0.45 kWh/kg$_{input}$, respectively. Bioethanol's productivity is 0.2 L per kg FORBI (Table 6).

**Table 6.** Inventory table, Scenario 3.

| Input/Output | Value |
|---|---|
| Waste Transportation to Landfill (km) | 25 |
| Fuel Consumption for collection (L/kg total wet weight) | $2.4 \times 10^{-3}$ |
| Fuel Consumption for transportation (L/kg total wet weight) | $8 \times 10^{-5}$ |
| Landfill gate fees (EUR/tonne) | 53 |
| Green waste collection and transportation (EUR/tonne) | 16 |
| FORBI production (tonne/tonneFU) | 0.08 |
| Sulphuric Acid for enzymatic hydrolysis (tonne/tonneFORBI) | $2 \times 10^{-2}$ |
| Enzymes for enzymatic hydrolysis (tonne/tonneFORBI) | $9 \times 10^{-3}$ |
| Heat for enzymatic hydrolysis (MJ/kginput) | 4.1 |
| Electricity for enzymatic hydrolysis (kWh/kginput) | $4.5 \times 10^{-1}$ |
| Bioethanol productivity (L/kgFORBI) | $2 \times 10^{-1}$ |

### 2.4.7. Scenario 4, Pellets

Pelletization is another alternative explored for the valorization of FORBI (Figure 7).

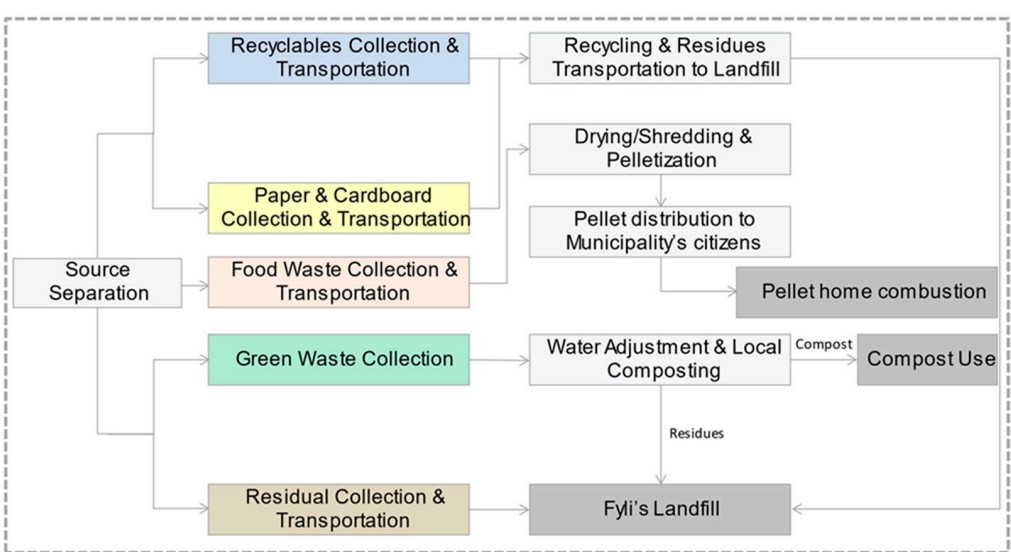

**Figure 7.** Scenario 4, pellets.

The bio-pellet produced is used to substitute conventional solid fuels in home pellet combustion devices. The electricity consumption during pelletization is equal to 0.0202 kWh per kg FORBI based on the Ecoinvent, while the substitution factor is $-0.77$ MJ$_{natural\ gas}$/MJ$_{pellet}$ based on fuels net calorific value (Table 7).

**Table 7.** Inventory table, Scenario 4.

| Input/Output | Value |
|---|---|
| Waste Transportation to Landfill (km) | 25 |
| Fuel Consumption for collection (L/kg total wet weight) | $2.4 \times 10^{-3}$ |
| Fuel Consumption for transportation (L/kg total wet weight) | $8 \times 10^{-5}$ |
| Landfill gate fees (EUR/tonne) | 53 |
| Green waste collection and transportation (EUR/tonne) | 16 |
| FORBI production (tonne/tonne$_{FU}$) | $8 \times 10^{-2}$ |
| Electricity consumption for pelletization (kWh/kg$_{FORBI}$) | $2.02 \times 10^{-2}$ |

*2.5. Uncertainty Analysis*

Consistent and balanced interpretation of LCA and LCC results require inclusion of an uncertainty analysis [46].

For the determination of input uncertainties, a pedigree matrix was developed, and lognormal distributions were acquired using the relevant Ecoinvent method [49]. The Ecoinvent method considers two kinds of parameter uncertainty, basic and additional. Basic uncertainty expresses the stochastic errors of the parameters due to, e.g., activity specific variability, while additional uncertainty is due to the use of imperfect data. Basic uncertainty is computed based on a table of factors provided by Ecoinvent, differentiated by exchange type and by class of process. Concerning additional uncertainty, the semi-quantitative scores obtained from the pedigree matrix are converted into uncertainty figures using additional uncertainty factors. The geometric standard deviation of the lognormal distribution is computed based on these uncertainty factors. Normal distributions were used for parameters acquired through experimental processes. The determination of normal distributions was conducted using Shapiro–Wilk test, in MS Excel. The special built-in module of EASETECH for conducting a Monte Carlo-based (10,000 calculations per scenario) uncertainty analysis was used.

It is important to highlight that most of the scenarios presented share common background data with the same uncertainty factors. Hence, the comparisons that will follow between the overall uncertainties of the scenarios should only be considered as indicative of the input data quality per scenario.

**3. Results & Discussion**

*3.1. Contribution Analysis*

3.1.1. Scenario 0

FW in the baseline scenario is not under a source separation scheme, but collected as commingled waste. The contribution analysis of Scenario 0 are presented in Figures 8 and 9 below.

Overall, Landfilling and Residual Collection and Transportation generates the most significant environmental and economic impacts.

The majority of the net impact for Climate Change and Ozone Depletion comes from the degradation of its organic matter in the landfill.

A significant net benefit for both Climate Change and Ozone Depletion comes from the recycling of dry recyclables that already takes place in the municipality. Concerning Climate Change, the results of the analysis show that the baseline scenario has an environmental impact of 65 kg $CO_2$-eq, while the contribution to the depletion of ozone is equal to $-7.6 \times 10^{-4}$ kg CFC-11 eq.

In the Human Toxicity (non-cancer effects) and Particulate Matter impact categories, the picture differs. Specifically, the largest percentage of the impact is a consequence of the recycling process of the municipality. On the other hand, landfilling seems to have a net positive impact in both categories. The total contribution of Scenario 0 to these two impact categories is $-4.7 \times 10^{-5}$ CTUh and $-4.3 \times 10^{-2}$ kg PM2.5-eq, respectively.

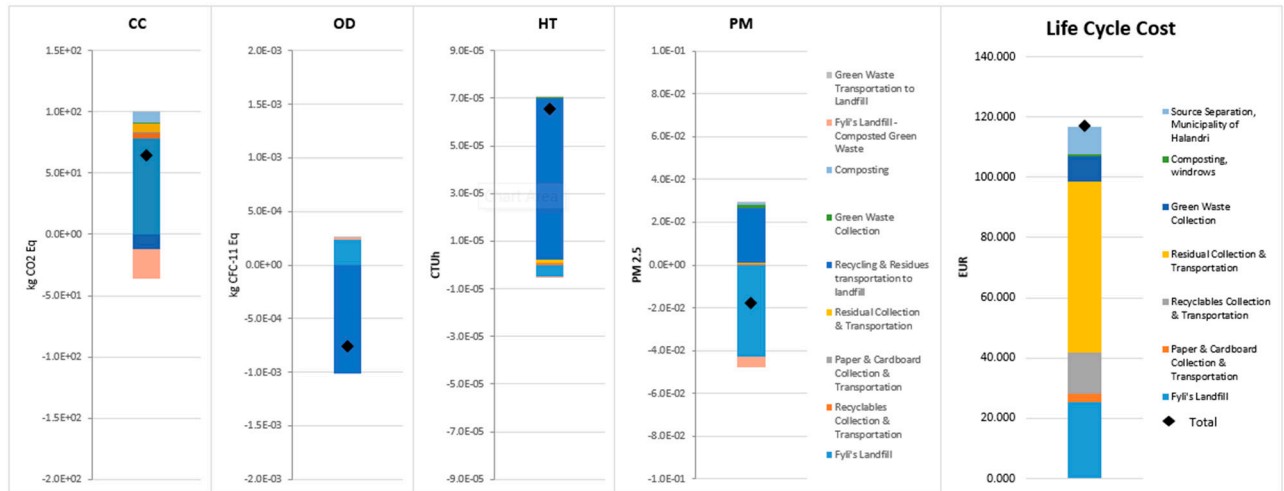

**Figure 8.** Scenario 0, LCA and LCC, contribution analysis and net results (presented as impacts' unit/ton MSW treated).

| Impact Category | CC | | OD | | HT | | PM | | LCC | |
|---|---|---|---|---|---|---|---|---|---|---|
| | kg CO2-Eq | Uncertainty | kg CFC-11 Eq | Uncertainty | CTUh | Uncertainty | kgPM2.5-eq | Uncertainty | EUR | Uncertainty |
| Fyli's Landfill | 7.8E+01 | 2.2E+00 | 2.4E-04 | 1.8E-06 | -4.7E-06 | 1.6E-07 | -4.3E-02 | 3.3E-04 | 2.5E+01 | 1.6E-01 |
| Recyclables Collection & Transportation | 4.0E+00 | 1.0E+00 | 1.5E-09 | 3.8E-10 | 8.1E-07 | 2.1E-07 | 5.6E-04 | 1.5E-04 | 1.4E+01 | 2.8E+00 |
| Paper & Cardboard Collection & Transportation | 1.0E+00 | 2.7E-01 | 3.8E-10 | 1.0E-10 | 2.1E-07 | 5.5E-08 | 1.4E-04 | 3.8E-05 | 2.8E+00 | 5.8E-01 |
| Residual Collection & Transportation | 6.8E+00 | 1.7E+00 | 2.5E-09 | 6.2E-10 | 1.4E-06 | 3.4E-07 | 9.4E-05 | 2.3E-04 | 5.7E+01 | 1.2E+01 |
| Recycling & Residues transportation to landfill | -1.2E+01 | 3.9E+00 | -1.0E-03 | 2.6E-05 | 6.8E-05 | 1.4E-06 | 2.6E-02 | 2.0E-03 | --- | --- |
| Green Waste Collection | 1.6E+00 | 5.7E-01 | 5.7E-10 | 2.1E-10 | 3.1E-07 | 1.1E-07 | 1.3E-03 | 4.6E-04 | 8.4E+00 | 9.8E-01 |
| Composting | 7.9E+00 | 4.2E-01 | 2.2E-10 | 2.7E-11 | 1.2E-07 | 1.4E-08 | 9.9E-04 | 5.2E-05 | 6.4E-01 | 2.4E-01 |
| Fyli's Landfill - Composted Green Waste | -2.4E+01 | 1.6E+00 | 2.1E-05 | 1.4E-06 | -5.2E-07 | 5.3E-08 | -5.0E-03 | 3.4E-04 | --- | --- |
| Green Waste Transportation to Landfill | 1.3E+00 | 4.7E-01 | 4.7E-10 | 1.7E-10 | 2.6E-07 | 9.5E-08 | 1.8E-04 | 6.5E-05 | --- | --- |
| Source Separation | --- | --- | --- | --- | --- | --- | --- | --- | 9.3E+00 | 4.9E-01 |
| **Sum** | **6.5E+01** | **5.7E+00** | **-7.6E-04** | **2.6E-05** | **6.6E-05** | **1.5E-06** | **-1.8E-02** | **2.3E-03** | **1.2E+02** | **1.7E+01** |

**Figure 9.** Scenario 0, LCA and LCC, contribution analysis.

Concerning the baseline scenario's cost, the processes that have the greatest impact are the collection and transportation of residual waste and the disposal of this waste in Fyli's Landfill.

Specifically, while the total cost of scenario 0 is computed equal to 120 € per tonne of waste, the cost for the collection and transportation of residual waste amounts to 57 € (49% of total cost). The disposal of residual waste requires an amount of 25 € per tonne of MSW due to gate fees that should be paid by the municipality for the amount of waste that ends up in landfills.

The LCIA figures presented indicate the significant importance of diverting food waste from landfills and developing a locally established—based on the principle of proximity, which indicates that MSW should be treated the closest possible to the generation area, instead of being transported to long-distance places—management and valorization paradigm.

The main factors/hotspots increasing both the LCIA (especially Climate Change) and the LCC of the baseline scenario are two:

1. The distance covered to transport food waste to the landfill significantly increases both the costs and the carbon footprint, basically because of the increased fuel consumption.
2. Additionally, food waste in landfill increases the gate fees paid by the municipality to a huge extent since food waste corresponds to more than 40% of the total MSW generation, as well as the GHGs emissions.

### 3.1.2. Scenario 0.1

Food waste source separation and the valorization of food and green waste for the production of compost leads to a significant reduction of the environmental impact related

to Climate Change. Specifically, Scenario 0.1 shows a net benefit equal to 34 kg $CO_2$ eq. The process of landfilling contributes most at that benefit. This is due to the fact that the disposed carbon is stabilized and can be considered as storage, having no interaction with the surrounding environment (benefit equal to 110 kg $CO_2$ eq).

Regarding Ozone Depletion, there is also an increase in net benefit compared to baseline scenario. This increase is attributed to the reduction of emissions during the collection, transportation and the landfilling of residual waste as a result of food waste source separation. The environmental benefit for the specific impact category equals to $9 \times 10^{-4}$ kg CFC-11 eq.

With respect to the human toxicity (non-cancer effects) and particulate matter impact categories, an increase of environmental burden is observed in comparison to Scenario 0. Specifically, the net benefit impact that is attributed to landfilling is reduced due to the reduction of residual waste that is disposed. The total contribution of Scenario 0.1 in these two impact categories is $7.1 \times 10^{-5}$ CTUh and $2 \times 10^{-2}$ kg PM2.5-eq, respectively. The contribution analysis of Scenario 0.1 is presented in Figures 10 and 11.

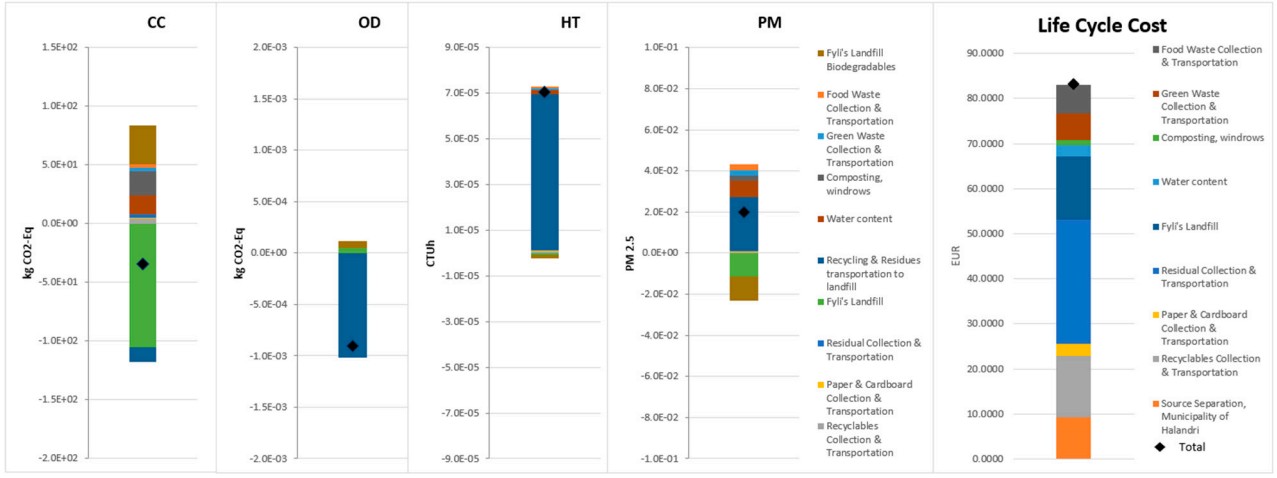

**Figure 10.** Scenario 0.1, LCA and LCC, contribution analysis and net results (presented as impacts' unit/ton MSW treated).

| Impact Category | CC | | OD | | HT | | PM | | LCC | |
|---|---|---|---|---|---|---|---|---|---|---|
| | kg CO2-Eq | Uncertainty | kg CFC-11 Eq | Uncertainty | CTUh | Uncertainty | kgPM2.5-eq | Uncertainty | EUR | Uncertainty |
| Recyclables Collection & Transportation | 4.0E+00 | 1.1E+00 | 1.5E-09 | 3.9E-10 | 8.1E-07 | 2.2E-07 | 5.6E-04 | 1.5E-04 | 1.4E+01 | 2.4E+00 |
| Paper & Cardboard Collection & Transportation | 1.1E+00 | 2.7E-01 | 3.9E-10 | 1.0E-10 | 2.1E-07 | 5.5E-08 | 1.5E-04 | 3.8E-05 | 2.8E+00 | 4.8E-01 |
| Residual Collection & Transportation | 3.0E+00 | 7.9E-01 | 1.1E-09 | 2.9E-10 | 6.1E-07 | 1.6E-07 | 4.2E-04 | 1.1E-04 | 2.8E+01 | 6.1E+00 |
| Fyli's Landfill | -1.1E+02 | 6.3E+00 | 4.9E-05 | 7.2E-06 | -8.8E-07 | 1.7E-07 | -1.1E-02 | 1.3E-03 | 1.4E+01 | 5.7E-01 |
| Recycling & Residues transportation to landfill | -1.2E+01 | 3.9E+00 | -1.0E-03 | 2.5E-05 | 6.8E-05 | 1.3E-06 | 2.6E-02 | 2.0E-03 | --- | --- |
| Water content | 1.6E+01 | 4.1E-01 | 1.4E-06 | 3.5E-08 | 2.0E-06 | 5.0E-08 | 8.2E-04 | 2.1E-04 | 2.4E+00 | 1.1E+00 |
| Composting, windrows | 2.0E+01 | 1.2E+00 | 4.0E-10 | 5.0E-11 | 2.1E-07 | 2.7E-08 | 2.5E-03 | 1.5E-04 | 1.2E+00 | 4.5E-01 |
| Green Waste Collection & Transportation | 2.8E+00 | 7.2E-01 | 1.0E-09 | 2.7E-10 | 5.6E-07 | 1.5E-07 | 2.3E-03 | 5.8E-04 | 5.8E+00 | 9.4E-01 |
| Food Waste Collection & Transportation | 2.6E+00 | 9.0E-01 | 1.3E-09 | 3.3E-10 | 7.3E-07 | 1.8E-07 | 2.9E-03 | 7.3E-04 | 6.4E+00 | 9.7E-01 |
| Fyli's Landfill Biodegradables | 3.3E+01 | 7.4E+00 | 6.4E-05 | 8.4E-06 | -1.5E-06 | 2.1E-07 | -1.2E-02 | 1.5E-03 | 0.0E+00 | 0.0E+00 |
| Source Separation | --- | --- | --- | --- | --- | --- | --- | --- | 9.3E+00 | 4.8E-01 |
| **Sum** | **-3.4E+01** | **2.3E+01** | **-9.0E-04** | **4.1E-05** | **7.1E-05** | **2.6E-06** | **2.0E-02** | **6.7E-03** | **8.3E+01** | **1.3E+01** |

**Figure 11.** Scenario 0.1, LCA and LCC, contribution analysis.

Carbon sequestration through the disposal of biostabilized (composted) food waste leads to a significant improvement in CC impact. There is also a significant decrease in Life Cycle Cost in comparison with the baseline scenario, due to decreased gate fees.

Finally, as regards Scenario's 0.1 Life Cycle Costing analysis, there is an important decrease in total cost in comparison with baseline scenario, which equals 29% (EUR 83 per ton waste).

The process that contributes the most in the total cost still remains the collection and transportation of residual waste, but the respective amount has been reduced significantly (EUR 28). The same results are observed for the process of landfilling as well.

### 3.1.3. Scenario 1.1

The anaerobic digestion of food waste and the composting of the effluent alongside with green waste that is implemented in Scenario 1.1 results in a decrease of the total kg $CO_2$-eq of the scenario. Specifically, there is a net benefit regarding Climate Change that equals $-49$ kg $CO_2$-eq. This environmental benefit is observed mainly due to the storage of stabilized carbon during landfilling and due to the utilization of biogenic carbon for the production of high added value products such as the compressed natural gas.

Regarding Ozone Depletion, there is an increase of the environmental benefit of the scenario compared to both baseline scenario and Scenario 0.1. The recycling process and the use of compost as a substitution of chemical fertilizers contributes to the mitigation of Ozone Depletion to a total of $-9.6 \times 10^{-4}$ kg CFC-11 eq.

As for the impact category "Human toxicity, non-cancer effects", there is an enhanced performance compared to Scenario 0.1, without however reaching the performance of Scenario 0. This decrease in the total net impact is due to the substitution of conventional fuel of waste collection trucks with the compressed natural gas that is produced. The total impact of Scenario 1.1 concerning toxic substances with non-cancer effects equals to $6.9 \times 10^{-5}$ CTUh.

Regarding Particulate Matter, there is a significant increase of the net impact of Scenario 1.1 ($5.6 \times 10^{-2}$ kg PM2.50-eq in total). The process of drying/shredding of FORBI is mainly responsible for this result. This process is really energy-intensive, and considering that the Greek energy mix is to a great extent based on lignite combustion, it contributes to the increase of the particulate matter of the scenario by $1.7 \times 10^{-2}$ kg PM2.5-eq. The contribution analysis for Scenario 1.1 is presented in Figures 12 and 13.

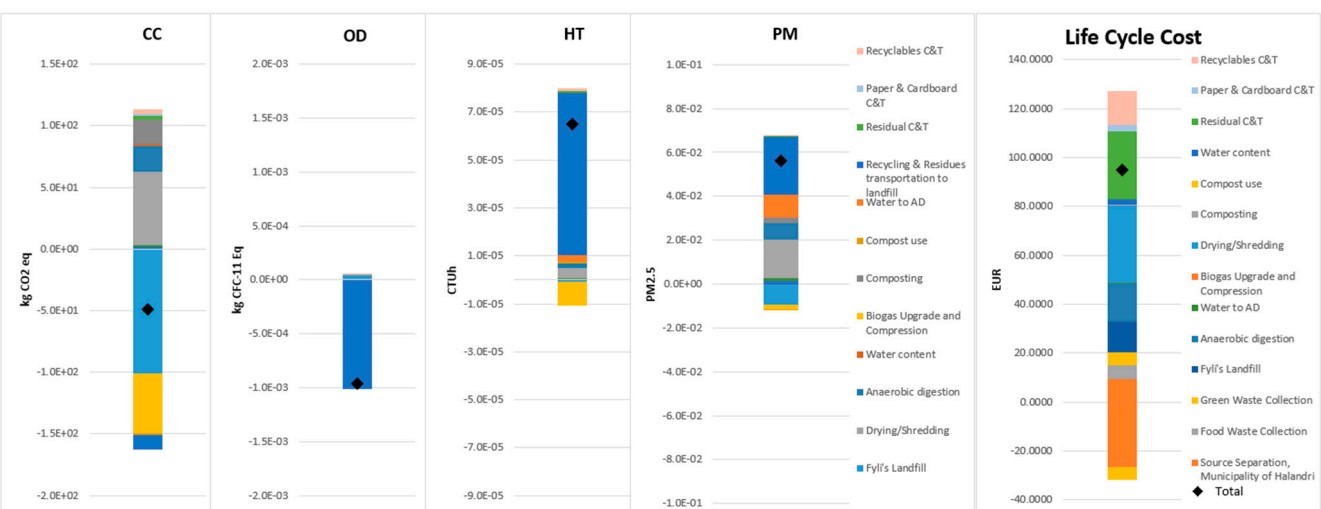

**Figure 12.** Scenario 1.1, LCA and LCC, contribution analysis and net results (presented as impacts' unit/ton MSW treated).

Landfilling of non-biodegradable waste and conventional fuel substitution lead to the most significant environmental benefits, while drying/shredding generate environmental impacts and costs, mainly due to fuel consumption.

The cost of the Scenario 1.1 is equal to EUR 95 per tonne of waste. An increase is observed in comparison to Scenario 0.1 due to the cost required for the equipment, the installation and the operation of drying/shredding and anaerobic digestion processes (EUR 31 and EUR 16, respectively). This extra cost that occurs for the required equipment

is partially balanced with the savings from the substitution of conventional fuel with CNG and of chemical fertilizers with the compost that is produced (EUR −27 and EUR 5).

The substitution of diesel and fertilizers by bio-CNG and compost, respectively, plays a significant role in the benefits for most of the impact categories of Scenario 1.1. This limits the replicability of the results to other municipalities since there might be no interest—or rationale—to do so.

| Impact Category | CC | | OD | | HT | | PM | | LCC | |
|---|---|---|---|---|---|---|---|---|---|---|
| | kg CO2-Eq | Uncertainty | kg CFC-11 Eq | Uncertainty | CTUh | Uncertainty | kgPM2.5-eq | Uncertainty | EUR | Uncertainty |
| Food Waste Collection | 2.0E+00 | 7.1E-01 | 7.3E-10 | 2.6E-10 | 4.0E-07 | 1.4E-07 | 1.6E-03 | 5.8E-04 | 5.7E+00 | 9.7E-01 |
| Green Waste Collection | 1.5E+00 | 5.5E-01 | 5.7E-10 | 2.0E-10 | 3.1E-07 | 1.1E-07 | 1.2E-03 | 4.5E-04 | 5.2E+00 | 9.2E-01 |
| Fyli's Landfill | -1.0E+02 | 7.2E+00 | 4.0E-05 | 7.5E-06 | -6.5E-07 | 1.8E-07 | -9.3E-03 | 1.3E-03 | 1.3E+01 | 5.8E-01 |
| Drying/Shredding | 6.0E+01 | 2.5E+00 | 1.1E-05 | 5.1E-07 | 4.1E-06 | 1.9E-07 | 1.7E-02 | 7.8E-04 | 3.1E+01 | 1.1E+01 |
| Anaerobic digestion | 2.1E+01 | 1.6E+00 | 7.6E-07 | 6.3E-08 | 2.0E-06 | 2.0E-07 | 7.6E-03 | 7.7E-04 | 1.6E+01 | 8.5E+00 |
| Water content | 1.4E+00 | 5.2E-02 | 1.5E-07 | 5.4E-10 | 3.3E-08 | 1.2E-09 | 2.5E-05 | 9.2E-06 | 2.2E+00 | 1.1E-01 |
| Biogas Upgrade and Compression | -4.8E+01 | 5.6E+00 | 4.5E-07 | 5.4E-08 | -1.0E-05 | 1.2E-06 | -2.1E-03 | 3.2E-04 | -2.7E+01 | 3.2E+00 |
| Composting | 2.0E+01 | 1.2E+00 | 2.4E-10 | 3.2E-11 | 1.3E-07 | 1.7E-08 | 2.5E-03 | 1.5E-04 | 6.8E-01 | 2.5E-01 |
| Compost use | -1.3E+00 | 6.2E-02 | -4.4E-13 | 2.2E-14 | 7.0E-07 | 3.4E-08 | -1.4E-04 | 7.0E-06 | -5.2E+00 | 7.3E-01 |
| Water to AD | 18,24 | 4.9E-01 | 8.3E-07 | 2.2E-08 | 2.7E-06 | 7.2E-08 | 1.1E-02 | 2.8E-04 | 2.2E-01 | 1.2E-02 |
| Recycling & Residues transportation to landfill | -1.2E+01 | 3.5E+00 | -1.0E-03 | 1.7E-05 | 6.8E-05 | 1.4E-06 | 2.6E-02 | 1.7E-03 | --- | --- |
| Residual C&T | 3.0E+00 | 8.1E-01 | 1.1E-09 | 3.0E-10 | 6.1E-07 | 1.6E-07 | 4.2E-04 | 1.1E-04 | 2.8E+01 | 6.3E+00 |
| Paper & Cardboard C&T | 1.1E+00 | 2.8E-01 | 3.8E-10 | 1.0E-10 | 2.1E-07 | 5.7E-08 | 1.5E-04 | 4.0E-05 | 2.8E+00 | 6.1E-01 |
| Recyclables C&T | 4.0E+00 | 1.1E+00 | 1.5E-09 | 3.9E-10 | 8.1E-07 | 2.1E-07 | 5.6E-04 | 1.5E-04 | 1.4E+01 | 2.8E+00 |
| Source Separation | --- | --- | --- | --- | --- | --- | --- | --- | 9.3E+00 | 4.6E-01 |
| **Sum** | **-4.9E+01** | **2.6E+01** | **-9.6E-04** | **2.5E-05** | **6.9E-05** | **3.9E-06** | **5.6E-02** | **6.7E-03** | **9.5E+01** | **3.6E+01** |

**Figure 13.** Scenario 1.1, LCA and LCC, contribution analysis.

### 3.1.4. Scenario 1.2

With the exception of the impact category "Climate Change", Scenario 1.2 shows similar results as Scenario 1.1 (Figures 14 and 15).

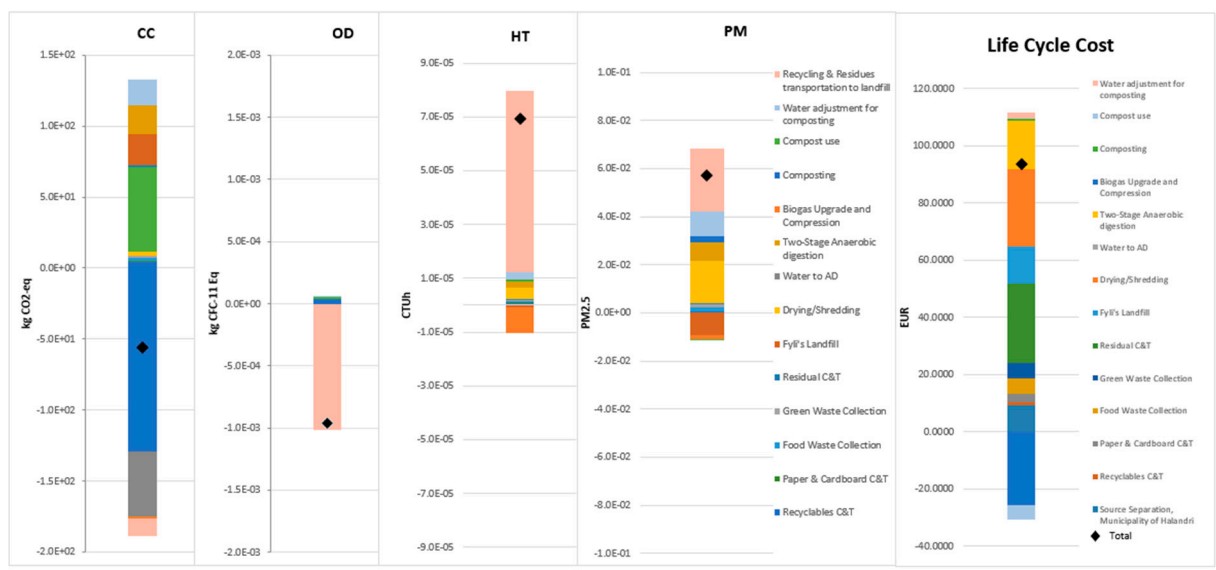

**Figure 14.** Scenario 1.2, LCA and LCC, contribution analysis and net results (presented as impacts' unit/ton MSW treated). Scenario 1.2's performance is comparable to Scenario 1.1.

Specifically, for the category of Climate Change, the current scenario has a net benefit equal to −56 kg $CO_2$ eq, which is the second greatest among all the alternative scenarios that were examined. This environmental benefit is mostly attributed to the carbon that is sequestered as storage during the landfilling process. This storage of carbon is equivalent to a saving of 130 kg $CO_2$ eq (22% increase compared to Scenario 1.1). On the contrary, the energy intensive process of drying/shredding sets an environmental burden regarding the benefit that could be derived from this scenario, contributing to the total net impact by 59 kg $CO_2$ eq.

| Impact Category | CC | | OD | | HT | | PM | | LCC | |
|---|---|---|---|---|---|---|---|---|---|---|
| | kg CO2-Eq | Uncertainty | kg CFC-11 Eq | Uncertainty | CTUh | Uncertainty | kgPM2.5-eq | Uncertainty | EUR | Uncertainty |
| Recyclables C&T | 4.0E+00 | 9.8E-01 | 1.5E-09 | 3.6E-10 | 8.0E-07 | 2.0E-07 | 5.5E-04 | 1.4E-04 | 1.4E+01 | 2.8E+00 |
| Paper & Cardboard C&T | 1.0E+00 | 2.8E-01 | 3.8E-10 | 1.0E-10 | 2.1E-07 | 5.6E-08 | 1.4E-04 | 3.9E-05 | 2.8E+00 | 5.6E-01 |
| Food Waste Collection | 2.0E+00 | 7.5E-01 | 7.4E-10 | 2.7E-10 | 4.1E-07 | 1.5E-07 | 1.6E-03 | 6.0E-04 | 5.7E+00 | 1.1E+00 |
| Green Waste Collection | 1.5E+00 | 5.3E-01 | 5.6E-10 | 1.9E-10 | 3.1E-07 | 1.1E-07 | 1.2E-03 | 4.2E-04 | 5.2E+00 | 9.3E-01 |
| Residual C&T | 3.0E+00 | 7.9E-01 | 1.1E-09 | 2.9E-10 | 6.1E-07 | 1.6E-07 | 4.2E-04 | 1.1E-04 | 2.8E+01 | 6.6E+00 |
| Fyli's Landfill | -1.3E+02 | 7.9E+00 | 4.1E-05 | 7.5E-06 | -6.5E-07 | 1.8E-07 | -9.4E-03 | 1.3E-03 | 1.3E+01 | 6.0E-01 |
| Drying/Shredding | 5.9E+01 | 2.7E+00 | 1.1E-05 | 5.4E-07 | 4.1E-06 | 2.0E-07 | 1.7E-02 | 8.2E-04 | 2.7E+01 | 9.0E+00 |
| Water to AD | 1.4E+00 | 5.2E-02 | 1.5E-07 | 5.4E-09 | 3.3E-08 | 1.2E-09 | 2.5E-04 | 9.1E-06 | 2.2E-01 | 1.2E-02 |
| Two-Stage Anaerobic digestion | 2.2E+01 | 1.6E+00 | 9.1E-07 | 6.6E-08 | 2.2E-06 | 2.0E-07 | 7.8E-03 | 7.5E-04 | 1.7E+01 | 9.6E+00 |
| Biogas Upgrade and Compression | -4.5E+01 | 4.3E+00 | 4.5E-07 | 4.6E-08 | -9.7E-06 | 9.2E-07 | -1.8E-03 | 2.6E-04 | -2.6E+01 | 2.6E+00 |
| Composting | 2.0E+00 | 1.2E+00 | 2.4E-10 | 3.0E-11 | 1.3E-07 | 1.6E-08 | 1.6E-03 | 2.5E-04 | 6.8E-01 | 2.4E-01 |
| Compost use | -1.3E+00 | 7.2E-02 | -4.4E-13 | 2.5E-14 | 6.9E-07 | 3.9E-08 | -1.4E-04 | 8.2E-06 | -5.2E+00 | 7.5E-01 |
| Water adjustment for composting | 1.8E+01 | 4.6E-01 | 8.3E-07 | 2.1E-08 | 2.7E-06 | 6.8E-08 | 1.0E-02 | 2.7E-04 | 2.2E+00 | 1.1E-01 |
| Recycling & Residues transportation to landfill | -1.2E+01 | 3.9E+00 | -1.0E-03 | 2.5E-05 | 6.8E-05 | 1.4E-06 | 2.6E-02 | 2.1E-03 | 0.0E+00 | 0.0E+00 |
| Source Separation | --- | --- | --- | --- | --- | --- | --- | --- | 9.2E+00 | 4.6E-01 |
| **Sum** | **-5.6E+01** | **2.5E+01** | **-9.6E-04** | **3.3E-05** | **6.9E-05** | **3.7E-06** | **5.7E-02** | **7.0E-03** | **9.4E+01** | **3.5E+01** |

**Figure 15.** Scenario 1.2, LCA and LCC, contribution analysis.

Regarding the cost factor of Scenario 1.2, a total cost of 94 € per tonne has been estimated, which corresponds to an approximate decrease of 20% compared to the baseline scenario.

Drying/shredding corresponds to the highest percentage of the costs, while significant savings (26 € per tonne) are achieved through the substitution of conventional waste trucks' fuels with hythane. Scenarios 1.1 and 1.2 are comparable regarding the costs' factor.

Regarding the substitution assumption, the same limitations as in Scenario 1.1 apply here.

### 3.1.5. Scenario 2

Scenario 2 shows the greatest environmental impact among all scenarios that were examined, for the impact category of Climate Change (Figures 16 and 17).

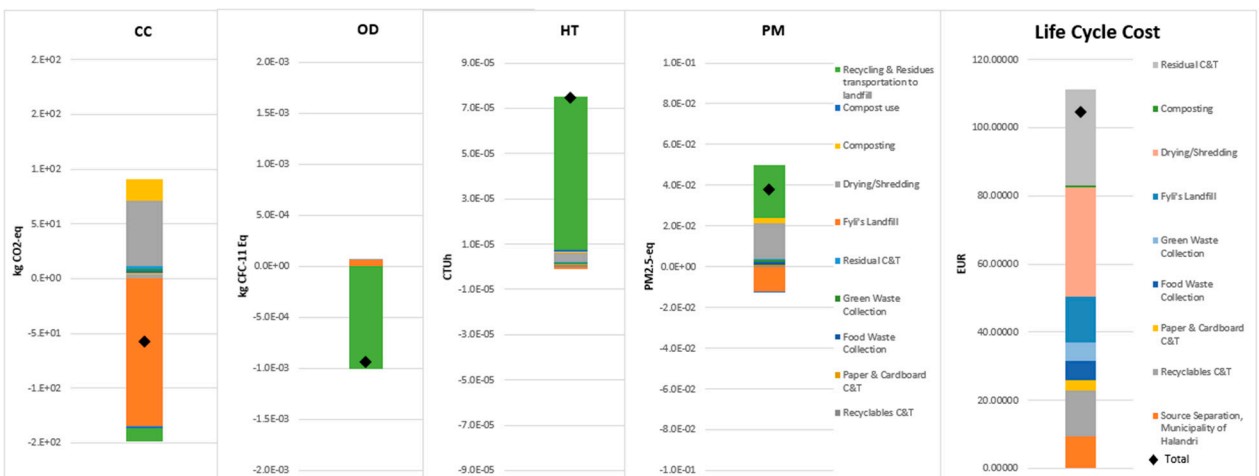

**Figure 16.** Scenario 2, LCA and LCC, contribution analysis and net results (presented as impacts' unit/ton MSW treated).

Food waste diversion from landfill and drying/shredding present the highest environmental and economic benefit and burden correspondingly, as in the previous alternative scenarios.

Specifically, it has a net benefit of −57 kg $CO_2$ eq per tonne of municipal solid waste that is treated. In that scenario too, the process that contributes the most to the reduction of the Climate Change impact category is the disposal of waste at Fyli's Landfill and consequently the storage of carbon (−130 kg $CO_2$ eq).

| Impact Category | CC | | OD | | HT | | PM | | LCC | |
|---|---|---|---|---|---|---|---|---|---|---|
| | kg CO2-Eq | Uncertainty | kg CFC-11 Eq | Uncertainty | CTUh | Uncertainty | kgPM2.5-eq | Uncertainty | EUR | Uncertainty |
| Recyclables C&T | 4.0E+00 | 1.0E+00 | 1.5E-09 | 3.8E-10 | 8.0E-07 | 2.1E-07 | 5.5E-04 | 1.4E-04 | 1.4E+01 | 2.5E+00 |
| Paper & Cardboard C&T | 1.1E+00 | 2.8E-01 | 3.9E-09 | 1.0E-10 | 2.1E-07 | 5.5E-08 | 1.5E-04 | 3.8E-05 | 2.9E+00 | 5.5E-01 |
| Food Waste Collection | 2.0E+00 | 7.1E-01 | 7.2E-10 | 2.6E-10 | 3.9E-07 | 1.4E-07 | 1.6E-03 | 5.8E-04 | 5.8E+00 | 1.1E+00 |
| Green Waste Collection | 1.5E+00 | 5.5E-01 | 5.7E-10 | 2.0E-10 | 3.1E-07 | 1.1E-07 | 1.3E-03 | 4.5E-04 | 5.2E+00 | 9.6E-01 |
| Residual C&T | 3.0E+00 | 7.8E-01 | 1.1E-09 | 2.9E-10 | 6.0E-07 | 1.6E-07 | 4.2E-04 | 1.1E-04 | 2.8E+01 | 6.3E+00 |
| Fyli's Landfill | -1.3E+02 | 8.4E+00 | 5.7E-05 | 6.7E-06 | -1.0E-06 | 1.7E-07 | -1.2E-02 | 1.2E-03 | 1.4E+01 | 5.3E-01 |
| Drying/Shredding | 6.0E+01 | 2.5E+00 | 1.1E-05 | 4.9E-07 | 4.1E-06 | 2.0E-07 | 1.7E-02 | 8.0E-04 | 3.2E+01 | 1.1E+01 |
| Composting | 2.0E+01 | 1.2E+00 | 2.7E-10 | 3.4E-11 | 1.4E-07 | 1.8E-08 | 2.5E-03 | 1.5E-04 | 7.8E-01 | 2.8E-01 |
| Compost use | -1.7E+00 | 6.6E-02 | -5.8E-13 | 2.3E-14 | 9.0E-07 | 3.6E-08 | -1.9E-04 | 7.5E-06 | -6.8E+00 | 8.0E-01 |
| Recycling & Residues transportation to landfill | -1.2E+01 | 4.0E+00 | -1.0E-03 | 2.6E-05 | 6.8E-05 | 1.4E-06 | 2.6E-02 | 2.1E-03 | --- | --- |
| Source Separation | --- | --- | --- | --- | --- | --- | --- | --- | 9.3E+00 | 4.6E-01 |
| Sum | -5.7E+01 | 2.0E+01 | -9.5E-04 | 3.3E-05 | 7.4E-05 | 2.5E-06 | 3.8E-02 | 5.5E-03 | 1.0E+02 | 2.5E+01 |

**Figure 17.** Scenario 2, LCA and LCC, contribution analysis.

Concerning Ozone Depletion, there is also observed an environmental benefit equal to $-9.5 \times 10^{-4}$ kg CFC-11 eq that. Landfilling of residual waste and the drying/shredding for the production of FORBI are the processes of the scenario that contributes most to the deterioration of the phenomenon.

As for the impact category "Human toxicity, non-cancer effect" there is an increase of 12% in comparison to the baseline scenario (net impact equals to $7.4 \times 10^{-5}$ CTUh). This deterioration of the phenomenon is mainly due to the process of drying/shredding, which contributes to the total toxic substances by $4.1 \times 10^{-6}$ CTUh. At the same time, the environmental savings that are attributed to the disposal of residual waste to Fyli's Landfill are reduced by 79% compared to the baseline scenario ($-1 \times 10^{-6}$ CTUh, while the net impact of the scenario 0 was $-4.8 \times 10^{-6}$ CTUh).

LCIA regarding particulate matter shows a total net impact equal to $3.8 \times 10^{-2}$ kg PM2.5 eq. The processes that contribute the most at this environmental impact are recycling and residues transportation to landfill and the energy intensive process of drying shredding with a contribution of $2.6 \times 10^{-2}$ kg PM2.5 eq and $1.7 \times 10^{-2}$ kg PM2.5 eq, respectively.

From the Life Cycle Costing analysis of Scenario 2, a total cost of EUR 1010 per tonne of municipal solid waste treated is computed, which is the highest among all alternative scenarios that are examined.

Compared to Scenario 0.1, there is an increase of total cost by 20% due to the cost of the drying/shredding process (EUR 32). At the same time, the savings attributed to the substitution of chemical fertilizers with the bio-based produced compost (EUR −6.8), are not enough to balance the total expenditure required for the production of FORBI. However, the total cost of the Scenario 2 remains lower than that of the baseline scenario due to reduced landfill gate fees and due to the local treatment of food waste.

The substitution of fertilizers by compost is responsible for a significant part of the Scenario's environmental and economic benefits. However, in this case (Scenario 2) this is not such a limiting factor—as in the previous scenarios—since there are multiple ways to utilize compost (commercially or for the municipal parks), hence it is an assumption more easily applicable to wider areas.

3.1.6. Scenario 3

Concerning the impact category of Climate Change for Scenario 3, there is also a net benefit for the system equal to −48 kg $CO_2$ eq (Figures 18 and 19).

The introduction of multiple bioprocesses (bio-EtOH production and anaerobic digestion) increases the monetary valorization cost, without correspondingly increasing the benefits. In terms of its environmental performance Scenario 3 performs similarly to Scenario 2, apart from HT and PM for which the corresponding impacts are doubled.

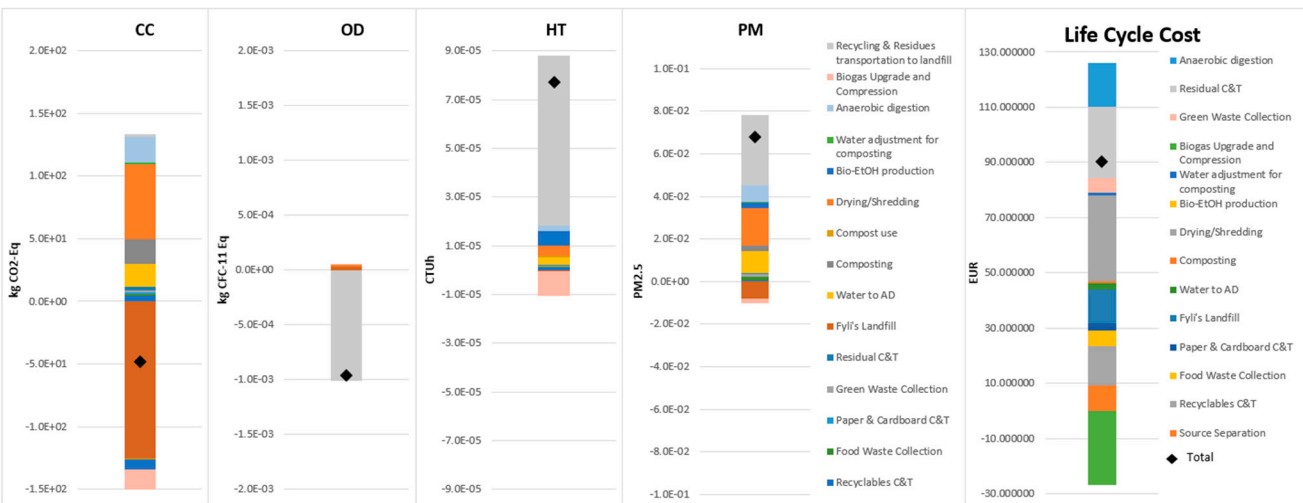

**Figure 18.** Scenario 3, LCA and LCC, contribution analysis and net results (presented as impacts' unit/tonneMSW treated).

| Impact Category | CC | | OD | | HT | | PM | | LCC | |
|---|---|---|---|---|---|---|---|---|---|---|
| | kg CO2-Eq | Uncertainty | kg CFC-11 Eq | Uncertainty | CTUh | Uncertainty | kgPM2.5-eq | Uncertainty | EUR | Uncertainty |
| Recyclables C&T | 4.1E+00 | 1.1E+00 | 1.5E-09 | 3.9E-10 | 8.3E-07 | 2.1E-07 | 5.7E-04 | 1.5E-04 | 1.4E+01 | 2.7E+00 |
| Food Waste Collection | 2.0E+00 | 7.0E-01 | 7.2E-10 | 2.6E-10 | 3.9E-07 | 1.4E-07 | 1.6E-03 | 5.6E-04 | 5.7E+00 | 1.0E+00 |
| Paper & Cardboard C&T | 1.0E+00 | 2.6E-01 | 3.8E-10 | 9.6E-11 | 2.1E-07 | 5.3E-08 | 1.4E-04 | 3.6E-05 | 2.8E+00 | 5.4E-01 |
| Green Waste Collection | 1.5E+00 | 5.7E-01 | 5.7E-10 | 2.1E-10 | 3.1E-07 | 1.1E-07 | 1.2E-03 | 4.6E-04 | 5.2E+00 | 9.0E-01 |
| Residual C&T | 2.9E+00 | 7.4E-01 | 1.0E-09 | 2.7E-10 | 5.7E-07 | 1.5E-07 | 4.0E-04 | 1.0E-04 | 2.6E+01 | 5.3E+00 |
| Fyli's Landfill | -1.3E+02 | 7.4E+00 | 3.4E-05 | 7.1E-06 | -5.0E-07 | 1.8E-07 | -8.1E-03 | 1.3E-03 | 1.2E+01 | 5.5E-01 |
| Water to AD | 1.8E+01 | 4.7E-01 | 8.3E-07 | 2.2E-08 | 2.7E-06 | 6.9E-08 | 1.1E-02 | 2.7E-04 | 2.2E+00 | 1.0E-01 |
| Composting | 2.0E+01 | 1.2E+00 | 2.4E-10 | 3.2E-11 | 1.3E-07 | 1.7E-08 | 2.5E-03 | 1.5E-04 | 6.9E-01 | 2.6E-01 |
| Compost use | -1.3E+00 | 7.9E-02 | -4.4E-13 | 2.7E-14 | 6.9E-07 | 4.3E-08 | -1.4E-04 | 8.9E-06 | 0.0E+00 | 0.0E+00 |
| Drying/Shredding | 5.9E+01 | 2.5E+00 | 1.1E-05 | 5.1E-07 | 4.1E-06 | 1.9E-07 | 1.7E-02 | 7.8E-04 | 3.1E+01 | 1.1E+01 |
| Bio-EtOH production | -8.0E+00 | 8.1E-01 | 1.9E-06 | 1.8E-07 | 6.3E-06 | 7.1E-07 | 2.8E-03 | 4.3E-04 | -8.0E+00 | 1.5E+00 |
| Water adjustment for composting | 1.4E+00 | 4.8E-02 | 1.5E-07 | 5.0E-09 | 3.3E-08 | 1.1E-09 | 2.5E-04 | 8.5E-06 | 2.2E-01 | 1.2E-02 |
| Anaerobic digestion | 2.1E+01 | 1.6E+00 | 7.7E-07 | 6.3E-08 | 2.0E-06 | 2.0E-07 | 7.6E-03 | 7.7E-04 | 1.6E+01 | 8.8E+00 |
| Biogas Upgrade and Compression | -4.7E+01 | 5.6E+00 | 4.4E-07 | 5.4E-08 | -1.0E-05 | 1.2E-06 | -2.1E-03 | 3.3E-04 | -2.7E+01 | 3.1E+00 |
| Recycling & Residues transportation to landfill | 2.1E+00 | 4.0E+00 | -1.0E-03 | 2.5E-05 | 7.0E-05 | 1.4E-06 | 3.3E-02 | 2.1E-03 | --- | ---- |
| Source Separation | ---- | ---- | ---- | ---- | ---- | ---- | ---- | ---- | 9.2E+00 | 4.4E-01 |
| **Sum** | **-4.8E+01** | **2.7E+01** | **-9.6E-04** | **3.3E-05** | **7.7E-05** | **4.7E-06** | **6.8E-02** | **7.4E-03** | **9.0E+01** | **3.6E+01** |

**Figure 19.** Scenario 3, LCA and LCC, contribution analysis.

Likewise, Scenarios 1.1 and 1.2, the two processes that contribute most to the total carbon footprint are the storage of carbon during landfilling of residual waste and the substitution of diesel at the municipal collection vehicles, offering an environmental benefit of $-130$ and $-47$ kg $CO_2$ eq, respectively. Conventional diesel enrichment with bioethanol as a substitution assumption is generally acceptable, since the relevant EU legislation predicts such an obligation for the future. On the other hand, the drying/shredding process burdens the environment with an impact of 59 kg $CO_2$ eq.

As for Ozone Depletion, there is an environmental benefit equal to $-9.6 \times 10^{-4}$ kg CFC-11 eq.

Regarding the impact categories of "Human toxicity, non-cancer effects" and "Particulate Matter" the results of the analysis show that scenario 3 has the greatest net impact among all waste management scenarios. Specifically, there is an environmental burden of $7.7 \times 10^{-5}$ CTUh and $6.8 \times 10^{-2}$ kg PM2.5 eq for these two categories. Compared to Scenarios 1.1 and 1.2, there is an increase of 11% regarding Human toxicity and of 16% regarding particulate matter. This increase is due to the production of bio-ethanol from FORBI. Bio-ethanol production sets an environmental burden equal to $6.3 \times 10^{-6}$ CTUh and $2.8 \times 10^{-2}$ kg PM2.5 eq, respectively.

Concerning costing analysis, the total cost of the Scenario 3 is EUR 90 per ton of waste. Compared to Scenarios 1.1 and 1.2, the cost is lower due to the use of the bioethanol that is produced, substituting 10% of conventional petrol in vehicles. The petrol substitution offers

savings equal to EUR −8, including the bioethanol production costs. The two other factors that contribute to savings of funds are the substitution of diesel in municipal collection trucks and the use of compost in the municipality's gardens instead of chemical fertilizers (EUR −27 and EUR −8, respectively). On the contrary, drying/shredding of food waste and the collection and transportation of residual waste to Fyli's Landfill represent an important share of the total cost of the scenario 3 (31 € and 26 €).

### 3.1.7. Scenario 4

Regarding Climate Change, Scenario 4 shows a net benefit equal to −28 kg $CO_2$-eq, which is the lowest among the six alternative waste management scenarios (Figures 20 and 21).

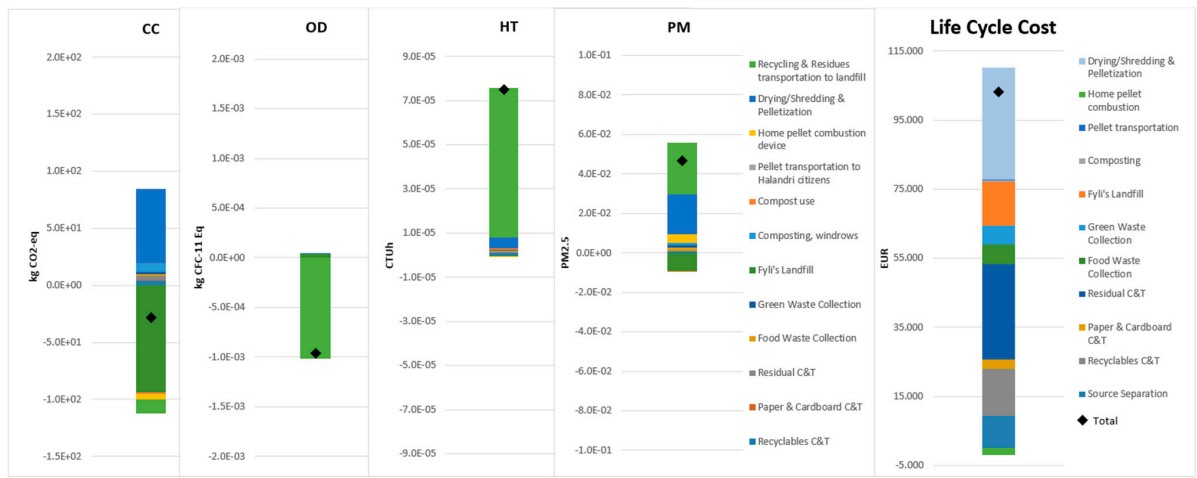

**Figure 20.** Scenario 4, LCA and LCC, contribution analysis and net results (presented as impacts' unit/ton MSW treated).

| Impact Category | CC | | OD | | HT | | PM | | LCC | |
|---|---|---|---|---|---|---|---|---|---|---|
| | kg CO2-Eq | Uncertainty | kg CFC-11 Eq | Uncertainty | CTUh | Uncertainty | kgPM2.5-eq | Uncertainty | EUR | Uncertainty |
| Recyclables C&T | 4.0E+00 | 1.0E+00 | 1.5E-09 | 3.7E-10 | 8.0E-07 | 2.0E-07 | 5.5E-04 | 1.4E-04 | 1.4E+01 | 2.7E+00 |
| Paper & Cardboard C&T | 1.0E+00 | 2.8E-01 | 3.8E-10 | 1.0E-10 | 2.1E-07 | 5.6E-08 | 1.4E-04 | 3.8E-05 | 2.8E+00 | 5.7E-01 |
| Residual C&T | 3.0E+00 | 8.4E-01 | 1.1E-09 | 3.1E-10 | 6.1E-07 | 1.7E-07 | 4.2E-04 | 1.2E-04 | 2.8E+01 | 6.1E+00 |
| Food Waste Collection | 2.0E+00 | 7.1E-01 | 7.2E-10 | 2.6E-10 | 4.0E-07 | 1.4E-07 | 1.6E-03 | 5.7E-04 | 5.7E+00 | 9.9E-01 |
| Green Waste Collection | 1.5E+00 | 5.3E-01 | 5.7E-10 | 2.0E-10 | 3.1E-07 | 1.1E-07 | 1.2E-03 | 4.3E-04 | 5.3E+00 | 9.4E-01 |
| Fyli's Landfill | -9.4E+01 | 6.1E+00 | 3.9E-05 | 7.1E-06 | -6.1E-07 | 1.7E-07 | -9.1E-03 | 1.3E-03 | 1.3E+01 | 5.9E-01 |
| Composting, windrows | 7.9E+00 | 5.2E-01 | 2.2E-10 | 2.8E-11 | 1.2E-07 | 1.5E-08 | 9.9E-04 | 6.4E-05 | 6.4E+00 | 2.3E-01 |
| Compost use | -1.2E+00 | 6.5E-02 | -4.3E-13 | 2.3E-14 | 6.7E-07 | 3.6E-08 | -1.4E-04 | 7.4E-06 | -5.1E+00 | 1.2E-01 |
| Pellet transportation to Halandri citizens | 1.0E-01 | 3.8E-02 | 3.8E-11 | 1.4E-11 | 2.1E-08 | 7.7E-09 | 8.3E-05 | 3.1E-05 | 2.4E-01 | 6.0E-02 |
| Home pellet combustion device | -4.9E+00 | 1.7E-01 | -1.8E-06 | 6.2E-08 | -5.0E-08 | 1.7E-09 | 4.3E-03 | 1.5E-04 | -1.9E+00 | 1.1E-01 |
| Drying/Shredding & Pelletization | 6.5E+01 | 2.7E+00 | 1.1E-06 | 5.3E-07 | 2.4E-06 | 1.2E-07 | 2.0E-02 | 8.7E-04 | 3.2E+01 | 1.1E+01 |
| Recycling & Residues transportation to landfill | -1.2E+01 | 3.8E+00 | -1.0E-03 | 2.4E-05 | 6.8E-05 | 1.3E-06 | 2.6E-02 | 2.0E-03 | --- | --- |
| Source Separation | --- | --- | --- | --- | --- | --- | --- | --- | 9.3E+00 | 4.6E-01 |
| **Sum** | **-2.8E+01** | **1.7E+01** | **-9.6E-04** | **3.2E-05** | **7.5E-05** | **2.4E-06** | **4.7E-02** | **5.6E-03** | **1.0E+02** | **2.4E+01** |

**Figure 21.** Scenario 4, LCA and LCC, contribution analysis.

Scenario 4 presents lower monetary cost and similar environmental performance compared to Scenario 3.

The drying/shredding process and the pelletization of the produced FORBI mitigate the environmental benefit, contributing to the total carbon footprint of the scenario by 65 kg $CO_2$ eq. Landfilling is—in that case too—the process with the greater benefit in terms of $CO_2$-eq, offering savings of −94 kg $CO_2$-eq, while there is also a benefit of −4.9 kg $CO_2$-eq, attributed to the use of pellets in home combustion devices, substituting natural gas.

As for Ozone Depletion, Scenario 4 shows the higher environmental benefit equal to $-9.6 \times 10^{-4}$ kg CFC-11 eq. Compared to the others alternative scenarios, this increase of savings in terms of kg CFC-11 eq, is due to the substitution of natural gas with bio-pellet produced from food waste ($-1.8 \times 10^{-6}$ kg CFC-11 eq).

Based on the results of the life cycle impact analysis for the category "Human toxicity, non-cancer effects" there is a total net impact of $7.5 \times 10^{-5}$ CTUh. Compared to Scenario 2,

although there is an environmental benefit from the substitution of natural gas with pellet, this benefit is outweighed due to high energy requirements of the drying/shredding and pelletization processes ($4.8 \times 10^{-6}$ CTUh). Moreover, Scenario 2's substitution assumption prerequisites that citizens will be willing to install the necessary equipment in their households to be able to use pellet as heat fuel, otherwise the generated pellet would be unused, thus eliminating the benefits of the scenario.

Regarding Particulate Matter, results of the analysis show a net impact equal to $4.7 \times 10^{-2}$ kg PM2.5-eq. The process with the highest contribution is drying/shredding and pelletization of FORBI with an impact of $2.0 \times 10^{-2}$ kg PM2.5-eq.

As regards Life Cycle Costing analysis of Scenario 4, a total cost of EUR 110 per tonne of waste treated is computed.

Drying/shredding and pelletization of FORBI alongside with the collection and transportation of residual waste to Fyli's Landfill are the two processes that represent the largest share of the total cost (EUR 32 and EUR 28, respectively). On the other hand, there are some savings that occur by the substitution of chemical fertilizers and natural gas with the high added value products produced from the valorization of green and food waste (EUR −5.1 and EUR −1.9).

### 3.2. Comparison of Alternative Scenarios

Figures illustrating the comparison of the alternative scenarios can be found in Figures 22–26 below.

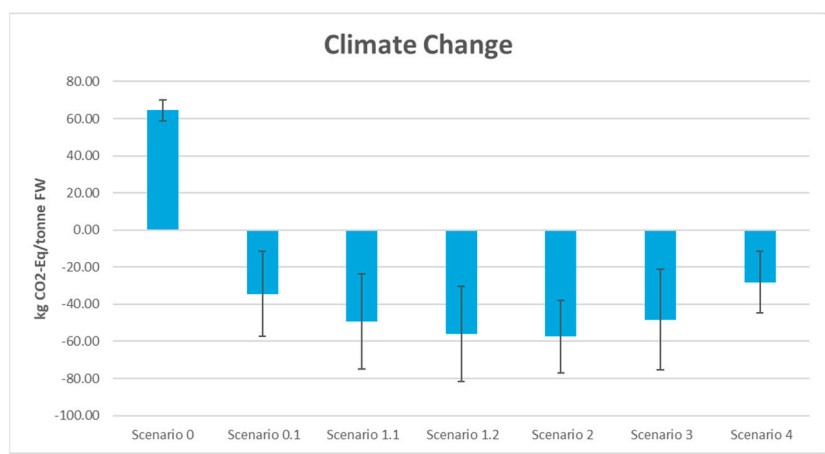

**Figure 22.** Comparison of alternative scenarios, Climate Change impact category.

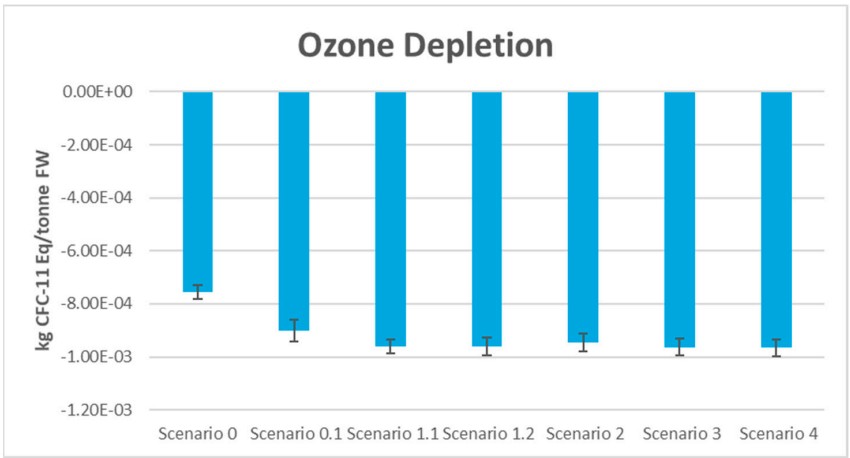

**Figure 23.** Comparison of alternative scenarios, Ozone depletion impact category.

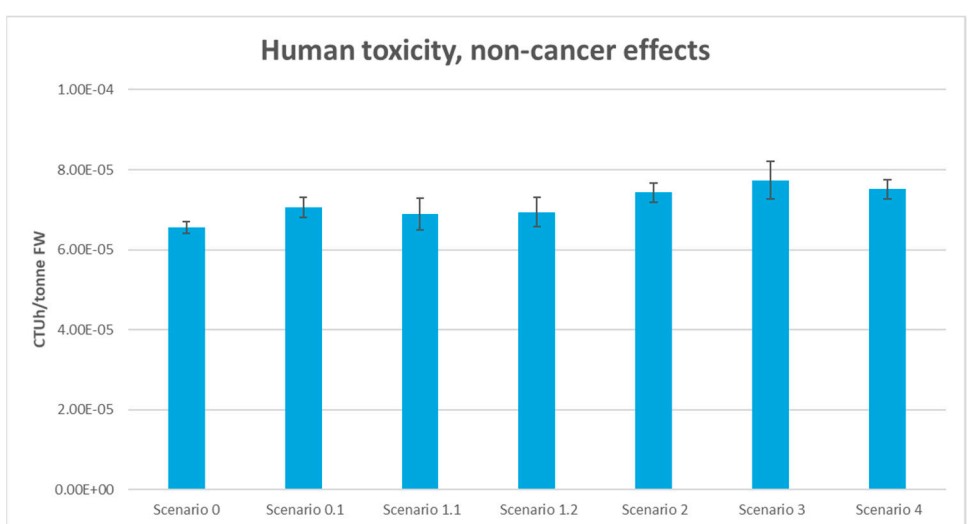

**Figure 24.** Comparison of alternative scenarios, Human toxicity, non-cancer effects.

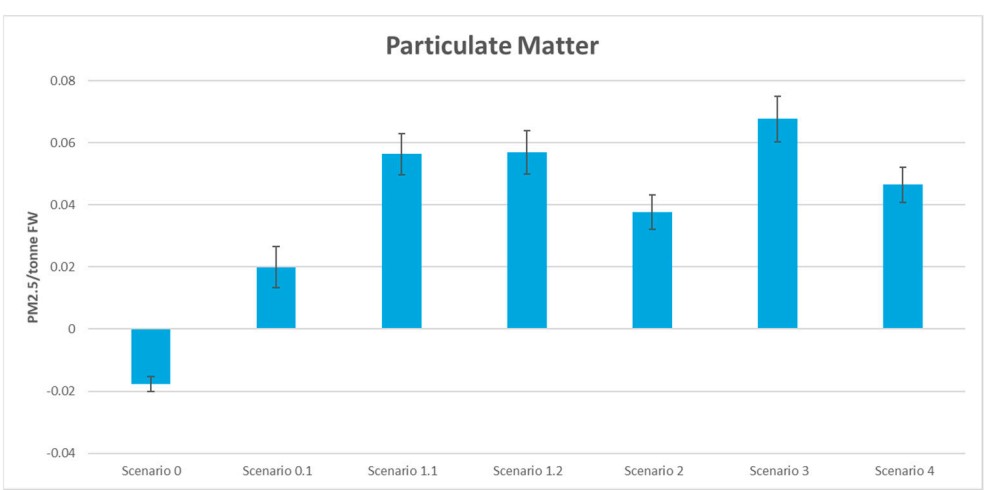

**Figure 25.** Comparison of alternative scenarios, Particulate Matter impact category.

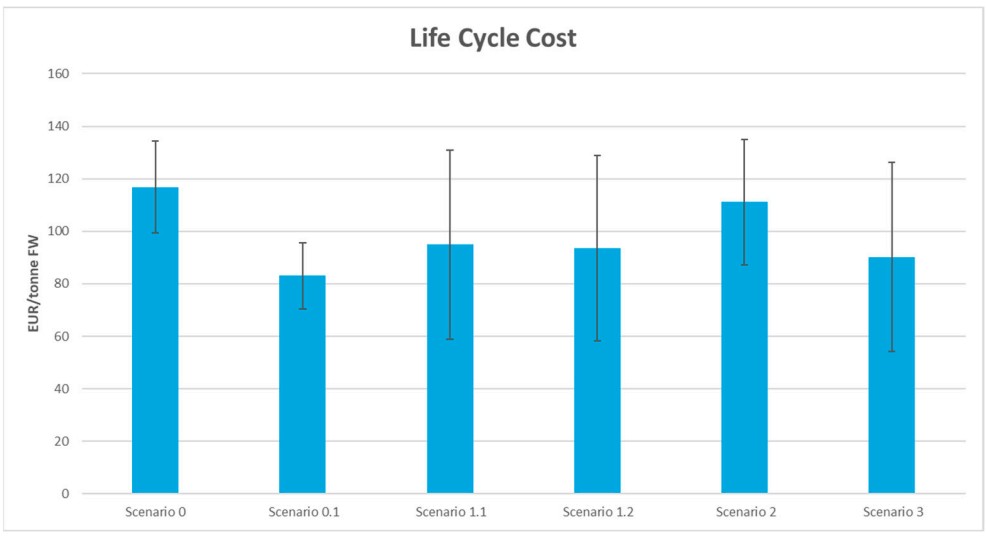

**Figure 26.** Comparison of alternative scenarios, Life Cycle Cost.

### 3.2.1. Climate Change (CC)

Based on the results of the uncertainty analysis that was conducted, for the impact category of climate change, it turns out that the baseline scenario has the least uncertain results with a range of $\pm 8.8\%$ (Figure 22).

Examining each process separately, it is observed that the processes that show the greatest variance in the results are the disposal of residual waste at Fyli's Landfill for the scenarios 1.1, 1.2 and 3 and right after the upgrading and compressing of biogas. As for scenarios 2 and 4, landfilling remains the process with the greatest variability, which follows the recycling of waste collected from blue and yellow bins. Finally, for the baseline scenario, the recycling process shows the highest uncertainty of results with a deviation of $\pm 3.9$ kg $CO_2$/tonne FW.

The baseline scenario leads to the highest CC burden, while all the alternative scenarios lead to net carbon benefits.

### 3.2.2. Ozone Depletion (OD)

Regarding Ozone Depletion, it is observed that all scenarios show much lower results' uncertainty in comparison to Climate Change (Figure 23).

Specifically, the uncertainty of the scenarios ranges from $\pm 2.6\%$ corresponding to Scenario 1.1, to $\pm 4.5\%$ for Scenario 0.1. For all waste management scenarios, the two processes that contribute most to the total uncertainty are recycling and landfilling of waste.

All the scenarios (including the current situation Scenario 0) lead to net environmental benefits regarding the Ozone Depletion impact category, with their overall performance being highly comparable.

### 3.2.3. Human Toxicity, Non-Cancer Effects (HT)

Concerning "Human toxicity, non-cancer effects" it is observed that all scenarios present low uncertainty (Figure 24). The uncertainty for this category ranges from $\pm 2.3\%$ (Scenario 0) to $\pm 6.1\%$ (Scenario 3). Recycling of waste collected from blue and yellow bins is—again—the process that shows the greatest fluctuation of results (Figure 24).

In the Human toxicity, non-cancer effects impact category, the baseline scenario (Scenario 0) seems to perform better compared to the other scenarios; however, the differences are not significant and fall below the uncertainty levels.

### 3.2.4. Particulate Matter (PM)

As for the Particulate Matter impact category, Scenario 0.1 shows the highest uncertainty, equal to 33.5% (Figure 25).

On the other hand, Scenario 3 seems to have the most precise results among all scenarios examined with an uncertainty of $\pm 11\%$. Examining processes separately, landfilling and recycling lead to the highest levels of uncertainty.

The baseline scenario performs better than the rest of the scenarios. However, in the case of the PMs, the differences are significant, with Scenario 0 leading to net environmental benefit, while the rest of the scenarios lead to significant net impact.

### 3.2.5. Life Cycle Cost (LCC)

Finally, regarding the Life Cycle Costing analysis, it is observed that Scenarios 0 and 0.1 present the least uncertain results (Figure 26).

Specifically, the cost of Scenario 0 ranges to 120 € $\pm$ 15%, while Scenario 0.1 ranges to 83 € $\pm$ 15%. The respective rate for Scenarios 4 and 2 is $\pm 22\%$. As for the highest rate of uncertainty, it corresponds to Scenario 3 with a percentage of $\pm 40\%$. Concerning the uncertainty of each process individually, for the baseline scenario and scenario 1, the collection and transportation of residual waste to Fyli's Landfill contributes most to the total uncertainty of the scenarios. Likewise, the process of drying/shredding shows the greatest variability of results for scenarios 1.1 to 4.

All the alternative scenarios lead to a net cost decrease, mostly due to the decrease of the transportation costs and the utilization of the various end-products.

*3.3. Policy Implications*

The LCA and LCC outcomes led to the identification of a number of principles that could be useful for local authorities and policymakers for the development of environmentally sound and economically feasible policies:

i.   The principle of proximity: For all the alternative scenarios, both from an economic and an environmental point of view, it was apparent that the long-distance transportation of waste to the landfill and/or the MRF led to the highest impact. Indeed, bringing the waste management sites closer to the municipalities (as in Scenarios 1.1. to 4) significantly decreased almost all the environmental impact categories and the overall waste management costs. Hence, it is important that policymakers focus on implementing waste management scenarios while also reflecting on the importance of waste transportation distances.

ii.  Food waste homogenization and deterioration: Drying and shredding food waste proved to be an efficient way of generating a homogenized, stable product suitable for a variety of alternative biological treatment options, leading to higher process efficiencies and better environmental and economic performance of the Scenarios.

iii. Food waste is the largest single municipal waste stream (in terms of quantities) and the least valorized. Hence, there is huge potential for implementing management and valorization approaches that will lead to waste diversion from landfills—hence greenhouse gases generation minimization—and generation of valuable resources and energy carriers.

Regarding the specific case of the Municipality of Halandri, the combination of the waste composition analysis data, the experimental data for the processes and the outcomes and conclusions of the LCA and LCC performed, it could be concluded that the food waste source separation scheme should be gradually expanded to cover the whole municipality. Furthermore, it is important that the municipality will focus on implementing suitable communication and awareness strategies to ensure maximum citizens participation and thus source separation rates. Irrespective of the specific valorization process to be implemented, the study's results demonstrated that the diversion of food waste from landfill will lead to significant environmental and economic benefits. Scenario 1.1 appeared as the most appropriate for the municipality both in terms of environmental and economic benefits and regarding the implementation simplicity compared to the other scenarios/technologies. The study demonstrated that it is important that the drying/shredding process will be further explored, aiming at minimizing its environmental and economic impacts (e.g., minimizing fuel consumption, investigation of potential alternative fuels like FORBI, etc.).

## 4. Discussion and Conclusions

The implementation of LCA and LCC tools proved that—in general—the baseline scenario performs significantly worse (both environmentally and economically) than the alternative scenarios. The use of experimental data to feed the models led to more accurate results in comparison with studies using generic/global data.

The simple alternative (Scenario 0.1) significantly decreases both the environmental and the economic footprint of the management scheme. However, it is apparent that it fails to maximize the environmental and economic benefits that food waste can offer due to its properties (organic content, nutrients, etc.).

Focusing on the critical parameters of the scenarios food waste transportation to the landfill significantly increases both the environmental footprint and the costs of the waste management approach. On the other hand, drying and shredding food waste for the generation of FORBI is proved to be a promising option, as it both leads to a homogenized (thus easier to handle) feedstock and leads to enhanced process efficiencies and consequently improved environmental and economic performance. However, it is

important that research focus will be put on minimizing its energy needs, which leads to the higher environmental and economic impacts, even though those are still lower compared to the baseline scenario.

The outcomes of the study were useful for the development of generic proposals that can be used as guiding principles by local authorities and policymakers in developing integrated and localized food waste valorization schemes.

The implementation of a substitution approach is associated with various sources of uncertainty since it is based on assumptions regarding future market conditions (supply and demand) that is impossible to ascertain. For example, Scenario 3 assumes the utilization of bioethanol as a partial substitute to vehicle fuels; however, this is an assumption that prerequisites the corresponding commercial willingness by the fuel producers and consumers. Furthermore, it should be noted that the system expansion approach is based on the region-specific data collected from the Municipality of Halandri. In case the scenarios presented in the current study were to be applied to another region, it should be done with caution, since a variety of factors (economic, social and environmental) might differ affecting the environmental and economic effectiveness of the technologies (e.g., lack of market for compost).

An important limitation of the current study is the exclusion of performance indicators other than economic and environmental. Specifically, environmental externalities as well as social aspects, such as human health, social cohesion and inclusion has not been considered in the assessed scenarios. Future research could build upon the findings of the current paper and integrate those aspects, leading to a more holistic view, and thus more accurate and effective policy making.

**Author Contributions:** Methodology, D.M., P.K., K.P. and G.L.; investigation, D.M., P.K., K.P. and G.L.; writing—original draft preparation, D.M., P.K. and K.P.; writing—review and editing, G.L. and T.F.A.; supervision, K.P. and G.L. All authors have read and agreed to the published version of the manuscript.

**Funding:** This work is produced under research project Horizon 2020, Grant Agreement No 688995. "Moving towards Life Cycle Thinking by integrating Advanced Waste Management Systems-[WASTE4THINK]".

**Institutional Review Board Statement:** Not applicable.

**Informed Consent Statement:** Not applicable.

**Data Availability Statement:** Not applicable.

**Conflicts of Interest:** The authors declare no conflict of interest. The funders had no role in the design of the study; in the collection, analyses, or interpretation of data; in the writing of the manuscript, or in the decision to publish the results.

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
