# Peer review of "Environmental and Economic Assessment of Alternative Food Waste Management Scenarios"

_sustainability, doi:10.3390/su14159634_

Round 1
Reviewer 1 Report
1. Author should be reconsidered the introduction on the literature of food waste but your title is the alternative Biodegradable Municipal Solid Waste management.
2. Autor should be stated the area of your study in title.
3. Author should be stated the reference on the line 101-109.
4. Author should be added the information of your sample area/country in introduction part.
5. Author should be added a table of comparison for information of all Scenarios.
6. Author should be added the calculation method or software utilization of LCA value.
7. Autor should be revised the discussion part with the update reference.
Author Response
Response to Reviewer 1 Comments
Point 1: Author should be reconsidered the introduction on the literature of food waste but your title is the alternative Biodegradable Municipal Solid Waste management.
Response 1: The title has been ammended according to the reviewer’s suggestion.
Point 2: Author should be stated the area of your study in title.
Response 2: The title has been revised accordingly.
Point 3: Author should be state the reference on the line 101-109
Response 3: Lines 101-109 include a statement made by the authors to summarize the conclusions that has been made through the literature review.
Point 4: Author should be added the information of your sample area/country in introduction part.
Response 4: Sample areas has been added in line 117.
Point 5: Author should be added a table of comparison for information of all Scenarios.
Response 5: Table 1 has been added in line 257 according to the reviewer’s suggestion
Point 6: Author should be added the calculation method or software utilization of LCA value.
Response 6: The software used (EASETECH) has been mentioned in line 130.
Point 7: Author should be revised the discussion part with the updated reference.
Response 7: See point 3 above
Reviewer 2 Report
This study well examined the environmental and economic performance of alternative Household Fermentable Waste management scenarios. The alternatives have been assessed using Life Cycle Assessment and Life Cycle Costing tools.
There are slight suggestions to be considered:
- Line 132, In Baseline data collection
It is highly recommended to explain the types of the collected /1000 kg data and its resources.
- Line 164, In life cycle Costing methodology Paragraph
The authors mentioned that only internal costs were included in the assessment.
Although, it was highly recommended to include the following types of costs:
1. Internal & External costs and
2. Budget costs, Externality costs, and Transfers.
Including these types of costs in the assessment would enrich the study.
- Line 170
The authors mentioned that only internal costs are included in the assessment (i.e., monetary costs)
It is highly recommended to specify which internal costs are included such as (Materials & energy consumption, labor costs, material, and energy sales, and Capital goods …)
Author Response
Response to Reviewer 2 Comments
Point 1: Line 132, in Baseline data collection: It is highly recommended to explain the types of the collected /1000 kg data and its resources
Response 1: A reference to another paper has been added, in which the whole baseline assessment methodology is explained in detail.
Point 2: Line 164, in Life Cycle Costing methodology Paragraph
The authors mentioned only internal costs were included in the assessment. Although, it was highly recommended to include the following types of costs:
- Internal & External costs and
- Budget costs, Externality costs and Transfers.
- Including these types of costs in the assessment would enrich the study
Response 2: The scope of the current study was to assess the LCC from the Municipality’s point of view. Hence, Externality costs and Transfers were not included.
Point 3: Line 170: The authors mentioned that only internal costs included in the assessment (i.e. monetary costs). It is highly recommended to specify which internal costs are included such as (Materials & energy consumption, labor costs, material and energy sales and Capital goods)
Response 3: The internal costs included has been mentioned in the Inventory Tables of each scenario.
Reviewer 3 Report
Dear Authors,
I find the current research of interest for the waste management sector towards circularity. Please find below, my minor suggestions:
1. From the point the view, in the paper title you should maintain eighter environmental and economic or Life Cycle Analysis and Life Cycle Costing assessment. In this state, in the title, similar information is provided twice.
2. Explain the abbreviations as their first appearance in the text (e.g. bio-CNG=bio- compressed natural gas; same for line 70 GHGs emissions, line 92 Life Cycle Assessment and Life Cycle Costing, for LCI and LCIA and ILCD
3. Since the research outputs are thanks to the Waste4think project, I recommend adding its website coordinates.
4. Equation 1. I think that you have intended LCC instead of CC
5. Table 4 . Please check the spelling of Comport production
Best regards
Author Response
Response to Reviewer 3 Comments
Point 1: From the point the view, in the paper title you should maintain eighter environmental and economic or Life Cycle Analysis and Life Cycle Costing assessment. In this state, in the title, similar information is provided twice.
Response 1: The title has been amended according to the reviewer’s suggestion.
Point 2: Explain the abbreviations as their first appearance in the text (e.g. bio-CNG=bio- compressed natural gas; same for line 70 GHGs emissions, line 92 Life Cycle Assessment and Life Cycle Costing, for LCI and LCIA and ILCD
Response 2: Abbreviations have been explained according to the reviewer’s suggestion.
Point 3: Since the research outputs are thanks to the Waste4think project, I recommend adding its website coordinates.
Response 3: Added in Line 15
Point 4: Equation 1. I think that you have intended LCC instead of CC
Response 4: Corrected, thank you
Point 5: Table 4 . Please check the spelling of Comport production
Response 5: changed to “Compost”
Round 2
Reviewer 1 Report
Accept for publication